# NANP targeting radiosensitizes glioblastoma through TNFR1 sialylation-driven mesenchymal shift

Yingwen Ding[1,2,3,6], Ze-Yan Zhang [1,2,3,6], Ravesanker Ezhilarasan[2,3], Aram S. Modrek [2,3,4], Melanie Graciani[2,3], Jerome Karp[2,3], Graysen McManus[2,3], Ananya Jambhale[2,3] & Erik P. Sulman [2,3,5] ✉

Glioblastoma (GBM) patients have dismal survival due to resistance to initial ionizing radiation therapy (RT). Clonal evolution analysis reveals no dominant RT-resistant clones, prompting a genome-wide CRISPR screen to identify radiosensitizing targets. The screening highlights DNA damage response genes, validating the effectiveness of our approach. N-acylneuraminate-9-phosphatase (NANP), a critical enzyme in the sialic acid synthetic pathway, is top-ranked in the screening and associated with patient outcomes. After radiation, NANP-deficient cells exhibit more DNA damage, G2/M arrest and apoptosis, and impaired DNA repair by favoring non-homologous end-joining over homologous recombination. Mechanistically, NANP influences NF-κB signaling and the mesenchymal state by modulating sialylation and inter-nalization of tumor necrosis factor receptor 1 (TNFR1), thereby affecting RT sensitivity. Intracranial orthotopic xenograft experiments validate the function of NANP in vivo. Here, we identify NANP as a radiosensitizing target dependent on TNFR1 sialylation and mesenchymal shift, providing a basis for developing RT sensitizers for GBM.

Glioblastoma (GBM) is the most common and aggressive adult malignant brain tumor with a median survival of about 15-17 months. Current standard of care for the treatment of patients with GBM involves maximal safe surgical resection of the tumor followed by concurrent chemotherapy with temozolomide (TMZ) and radiotherapy (RT) at a dose of 60 Gy in 30 fractions daily, and then adjuvant TMZ[1,2]. Addition of radiation alone improved survival over surgery by 6 months in the Brain Tumor Study Group trials[3–5]. Subsequent addition of TMZ improved survival an additional 2 months and, recently, addition of tumor treating fields further improved survival another 2 months[1,2,6]. Despite the potential benefits that each of these advances has, RT continues to be the most impactful first-line treatment of GBM with respect to survival, and thus, resistance to RT serves as a major factor in disease progression.

Nearly all patients succumb to tumor recurrence, hypothesized to be due to the presence of GBM stem-like cells (GSCs), a subpopulation within the tumor with self-renewal and therapy-resistant properties[7,8]. Given the benefit of RT in the treatment of GBM, attempts have been made to increase dose to the tumor, however, these approaches have failed to improve survival due to the resulting increased toxicity[9]. Therefore, there is an urgent need to develop strategies to enhance

[1]School of Basic Medical Sciences, Institute of Biomedical Innovation, The MOE Basic Research and Innovation Center for the Targeted Therapeutics of Solid Tumors, Provincial Key Laboratory of Tumor Biology, Jiangxi Medical College, Nanchang University, Nanchang, China. [2]Department of Radiation Oncology, New York University (NYU) Grossman School of Medicine, New York, NY, USA. [3]Brain and Spine Tumor Center, Laura and Isaac Perlmutter Cancer Center, NYU Langone Health, New York, NY, USA. [4]Department of Radiation Oncology, Keck School of Medicine of University of Southern California (USC), Los Angeles, CA, USA. [5]Department of Radiation Oncology, Duke University School of Medicine, Durham, NC, USA. [6]These authors contributed equally: Yingwen Ding, Ze-Yan Zhang. ✉e-mail: erik.sulman@duke.edu

radiosensitivity of GBM, especially the GSC population, in order to improve patient outcomes.

Transcriptome analysis of TCGA (The Cancer Genome Atlas) samples have classified GBM tumors into three subtypes: Classical (CL), Proneural (PN), and Mesenchymal (MES)[10]. These states however exhibit plasticity and recent studies have further refined the extent of plasticity and heterogeneity at the single-cell level[11]. Our previous study demonstrated NF-κB activation promotes MES differentiation of GSCs and GSCs with a MES phenotype displays increased radioresistance[12]. Further study showed upregulation of N-cadherin, a mesenchymal marker, mediates adaptive radioresistance in GBM and protects from radiation induced apoptosis[13]. Thus, MES phenotype disruption has been an attractive approach for radiosensitization[14,15].

Sialic acid residues attached to cell surface glycans play a crucial role in cell adhesion, migration, and interaction with the environment[16,17]. Aberrant sialylation is implicated in cancer progression and therapeutic resistance[18]. N-acylneuraminate-9-phosphatase (NANP) functions in the final step of sialic acid synthesis. In this work, we report the identification of NANP as a key regulator of radiosensitivity in GSCs using genome-wide CRISPR screening. We find that NANP regulates NF-κB pathway activation and the MES phenotype by modulating sialylation of TNFR1, ultimately impacting the radiosensitivity of GBM.

## Results

### Clonal evolution during radio-resistance

We previously reported that a few drug-resistant clones can dominate targeted therapy resistance through clonal evolution[19], which suggesting opportunities to overcome resistance by targeting these dominant clones. We sought to undertake a similar approach with radiation to identify dominant resistant clones that could be targeted to improve radiosensitivity. To that end, we studied the clonal evolution of the patient-derived GSC line GSC20 in response to fractioned RT using our CAPTURE barcoding approach (Supplementary Fig. 1a). The clonal evolution pattern in response to RT was strikingly different to that seen in response to targeted therapy (Supplementary Fig. 1b and Supplementary Data 1). In our previous study with targeted therapy[19], specifically the mutant BRAF inhibitor vemurafenib in a melanoma model, the top ten most abundant lineages contributed an impressive 85.35% to the final resistant pool (49.27-fold higher enrichment than that seen in control). However, in the RT study, this was only 7.97% (2.23-fold higher enrichment than that seen in control). The top enriched barcoded lineages in each experimental replicate were further selected based on GFOLD analysis (Supplementary Fig. 1c), and these lineages were rarely shared across experimental replicates (Supplementary Fig. 1d). Taken together, these results suggest that, unlike targeted therapy, RT resistance was not driven by a few dominant clones but instead may represent a stochastic selection, making it less likely to exploit a resistant clonal lineage.

### Genome-wide CRISPR screen for radiation sensitizing targets

Since the clonal lineage approach failed to identify targetable, dominant RT resistant clones and considering the toxicity of RT dose escalation, RT sensitization may serve as a better strategy to minimize RT resistance. Hence, we sought to identify RT sensitizing targets through a genome-wide CRISPR screen. We performed a genome-wide CRISPR screen to identify genes whose gRNAs drop-off in fractioned RT treated group compared to sham radiation group (Fig. 1a and Supplementary Data 2). GSC20 was used based on its marked resistance to RT. We transduced the cells with the Brunello lentiviral CRISPR library, which targets 19114 protein-coding genes (4 gRNAs per gene), at a multiplicity of infection (MOI) < 0.3. Experiments were performed as duplicates at 300-fold library representation per biologic replicate. After fractioned RT (2 Gy × 5 fractions) or sham treatment, cells underwent prolonged culture for additional 25 days. We then amplified gRNAs from genomic DNA for next generation sequencing. Dropout of individual genes was determined by multiple gRNAs targeting the same gene (Fig. 1b, c).

Gene set enrichment analysis (GSEA) of the ranked resulting gene list (Fig. 1d and Supplementary Fig. 2) demonstrated that non-homologous end-joining (NHEJ) and the related protein complexes (e.g., DNA ligase IV-XRCC4-XLF complex) were the most negatively enriched, followed by the enrichment of many other DNA damage response (DDR) pathways. These results were aligned with the well-established importance of DDR in response to radiation and under-score the overall effectiveness of the CRISPR screen. Using a filter based on a greater than 2-fold reduction in representation with a $p$-value of less than 0.05, as well as requiring at least two gRNAs targeting the same genes to ensure consistent outcomes, we identified 139 potential targets for RT sensitization. (Fig. 1e). We further interrogated these candidates to identify what pathways/functions were they involved by Gene Ontology (GO) and Reactome analyses. This revealed overrepresentation for genes involved in 11 major classes of pathways (Fig. 1f). "DNA damage response and repair" relating pathways were the most significantly enriched, and 7 of 10 other major classes of pathways, including "cell cycle checkpoints", "TP53 pathway", "DNA replication and recombination", "virus life cycle", "cell proliferation and differentiation", "post-translation protein modification", "transcription regulation", were not surprising to found enrichment given their related genes were highly overlapped with those involved in DDR. There were 23 genes with a direct function in key DNA double-strand break (DSB) repair pathways including 7 genes central to NHEJ (e.g., PRKDC, NHEJ1, DCLRE1C, XRCC4, LIG4, CYREN, HMGB2), 4 genes involved in homologous recombination (HR) (RAD51D, BRCA2, RAD17, RAD9A), and 12 genes central to general DDR (e.g., ATM, RNF8, RNF168, TP53BP1, TONSL, FOXM1, TDG, ERCC6L2, RFWD3, RAD1, TRAIP, TCEA1). Collectively and not surprisingly, these findings demonstrate that the genes involved in DDR pathways, particularly NHEJ, are major targets for increasing radiosensitivity in GSCs.

In addition to the these known DDR-related pathways, we also observed enrichment of genes involved in other pathways such as "nucleotide salvage", "sterol homeostasis" and "animo sugar metabolism" pathways. Given that nucleotides serve as the building blocks for DNA synthesis, it is unsurprising to find the "nucleotide salvage" pathway among these pathways. In fact, recently studies[20,21] have established its significance in regulating radiation sensitivity, further highlighting the success of the screen. The observation of the association between the "sterol homeostasis" pathway and RT sensitivity was somewhat unexpected. However, it is intriguing to note that several initial studies have indeed demonstrated its significant role in modulating RT sensitivity and DDR[22–24]. "Animo sugar metabolism" pathway was another unexpected RT sensitivity associated pathway, which has not been previously studied.

### Identification of NANP as a potential radiosensitizing target

To identify novel targets, we focused on the pathway "animo sugar metabolism" since this had not previously been known as involved in radiation sensitivity. Among the hits implicated in this pathway, NANP was chosen because of its high rank in our screening (Fig. 1c, g) and its association with GBM patient survival and tumor aggressiveness (Fig. 2a–h). NANP catalyzes the dephosphorylation of N-acetylneuraminic acid 9-phosphate (Neu5Ac-9P) to generate N-acetylneuraminic acid (Neu5Ac), the most common sialic acid in humans[17]. Interestingly, other NANP related pathways showed enrichment in our screen (Supplementary Fig. 2a), such as N-glycan biosynthesis.

The analysis of multiple public patient cohorts revealed that NANP expression was significantly higher in GBM samples compared to non-tumor controls (Fig. 2a and Supplementary Fig. 3a), as well as in comparison to oligodendroglioma, oligoastrocytoma, and

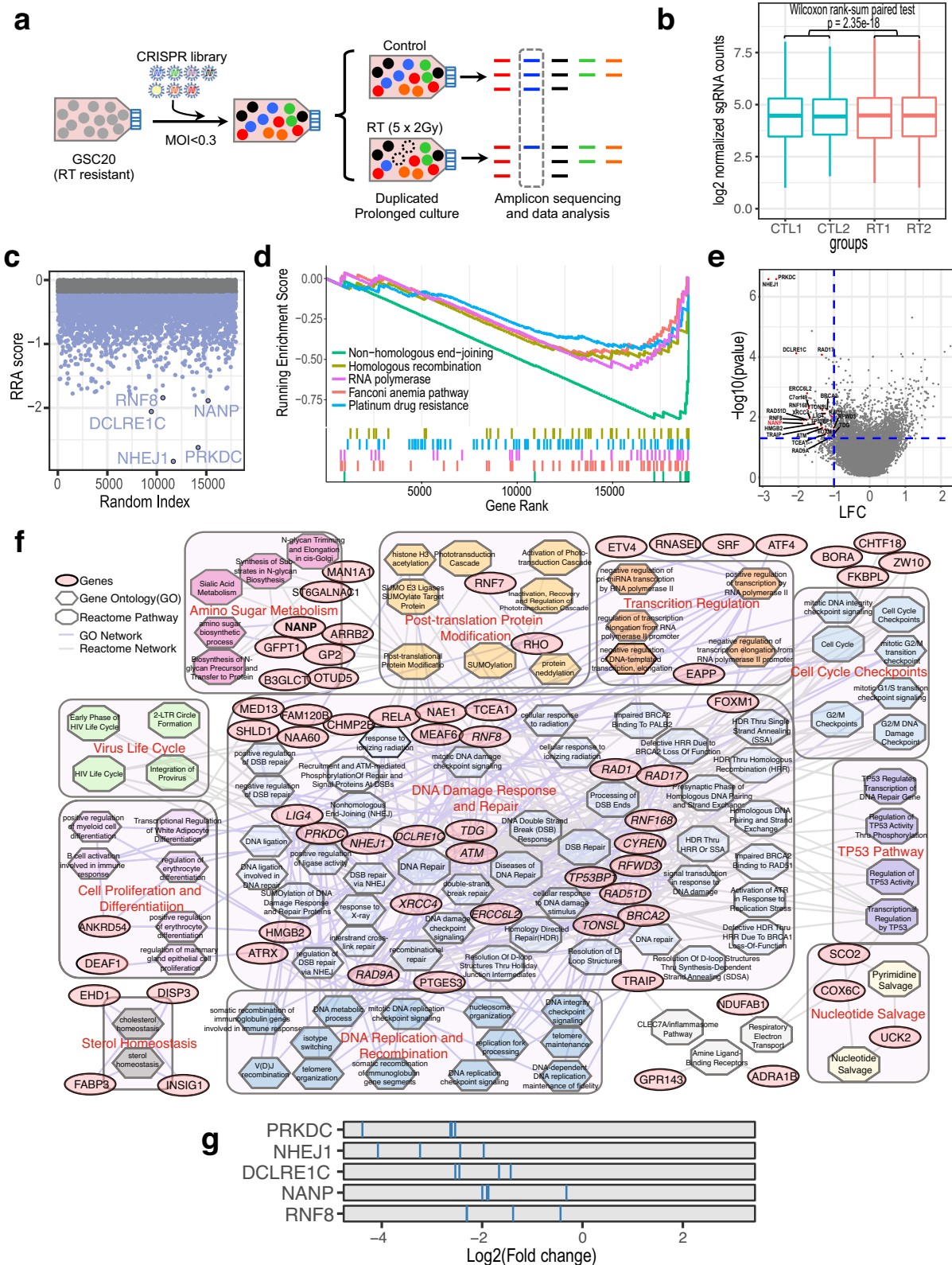

astrocytoma samples (Fig. 2b). There was also a notable elevation in *NANP* expression with increasing tumor grade (Fig. 2c, d and Supplementary Fig. 3b). Moreover, *NANP* expression was significantly higher in recurrent and secondary GBM when compared to the primary tumor (Fig. 2e). It was also significantly upregulated in GSCs, which are associated with RT resistance[7], compared to paired matched tumor from which each GSC was derived (Fig. 2f). Importantly, *NANP*'s high

expression was associated with poor patient survival in multiple cohorts (Fig. 2g, h and Supplementary Fig. 3c).

We also analyzed genes with colinear expression to NANP in The Cancer Genome Atlas (TCGA) dataset and identified a list of genes significantly ($p < 0.05$) correlated with *NANP* expression (Supplementary Data 3). Enrichment analyses were significant for the "N-glycan Synthesis" pathway (Fig. 2i, j), aligning with reported functions of

**Fig. 1 | Genome-wide CRISPR screen for radiation sensitizing targets.**
**a** Schematic outline of the experimental design for genome-wide CRISPR screen. Created in PowerPoint with elements created in BioRender. Zhang, Z. (2025) https://BioRender.com/gnecstk. **b** Box plot showing the distribution of log2 normalized sgRNA counts for different replicates, with and without RT treatment. Box plots show the median (center line), interquartile range (IQR, box: 25th-75th percentiles), and whiskers extending to the minimum/maximum values within 1.5×IQR. **c** Scatterplot showing robust rank aggregation (RRA) score ranking results of the negative selected genes from the CRISPR screen. The top 5 negative selected genes

are annotated. **d** GSEA enrichment plot showing the top 5 negative enrichment pathways of the CRISPR screen. **e** Volcano plot showing the CRISPR screen with the x-axis showing log2 fold change (LFC); the blue dashed lines showing the cut off lines for LFC < −1 and p-value < 0.05. The gene-level p-value was calculated used RRA with Wald test to aggregate sgRNA-level significance (modeled by negative binomial distribution), two-sided. **f** Network plot showing Gene Ontology and Reactome pathway network of the top 139 negative selected genes. **g** Rank plot of the sgRNAs ranking of the top 5 negative selected genes.

---

NANP in sialylation. Surprisingly, several key pathways related to DNA damage repair, including "mismatch repair," "DNA replication," "base excision repair," and "homologous recombination" were also significantly enriched (Fig. 2i, k), suggesting a potential role for NANP in DDR. These findings highlight the potential involvement of NANP in glycan synthesis processes and DDR, prompting us to further investigate its functional role in radiation response.

### Targeting NANP radiosensitizes GBM

To interrogate and validate the role of NANP in radiation response, we first performed clonogenic and cell viability assays after radiation treatment. GSCs (GSC20 and GSC11) were transduced with four independent NANP-targeting short hairpin RNAs (shRNAs) (shNANP-1/3/4/5) or non-targeting control (shCTL) through lentiviral transduction. Silencing efficiency ranged from 60% to 90% among these shRNAs (Supplementary Fig. 4a, b, 5a, b and 6a, b). NANP attenuation significantly improved radiation sensitivity in both GSC20 and GSC11 (Fig. 3a, b). The RT sensitizing effect of NANP silencing was also observed in the glioma cell line U87 (Supplemental Fig. 6c).

RT primarily exerts its cytotoxic effects by inducing DNA damage, especially DSB, which often triggers G2/M cell cycle arrest to allow time for DNA repair and, if repair fails, the cells undergo apoptosis[25]. To further investigate how NANP effects DDR, we accessed the radiation induced cell cycle arrest and apoptosis in GSCs with and without NANP silencing. We observed a higher proportion of NANP silenced cells are still in the G2/M 24 h after radiation compared to controls (Fig. 3c), similar to results seen with NHEJ1 silencing which was used as a positive control. We noted increased G2/M arrest with more NANP silencing across the range of shRNAs used. This observation was consistent across GSC20, GSC11, and the U87 cell lines with 2-4 different shRNAs (Supplemental Figs. 4c, 5c and 6d). Importantly, at 24 h post-radiation, we also observed increased apoptosis in both GSC20 and GSC11 cells with NANP silencing compared to control cells (Fig. 3d, e and Supplemental Fig. 5d, e). Additionally, a time course study in GSC20 cells showed that after 8 Gy radiation, apoptosis increased over time and NANP silenced cells had higher apoptotic rates compared to control cells at 24, 48, 72, and 96 h, (Supplemental Fig. 7). When exposed to a higher dose of 16 Gy, NANP silenced cells exhibited more apoptosis than that was induced by 8 Gy, while control cells showed similar levels of apoptosis compared to that induced by 8 Gy (Supplemental Fig. 7). Using CRISPR system, we also validated these findings in NANP knockout cells (Fig. 3f, g and Supplemental Fig. 8a, b). Moreover, re-expression of shNANP1-resistant NANP (NANPr), which expresses wild-type NANP but uses alternative codons based on wobble base pairing, rescued the radio-sensitizing phenotype of NANP silencing (Fig. 3h and Supplemental Fig. 8c). Together, these findings suggest that NANP silencing resulted in increased G2/M cell cycle arrest and subsequent apoptosis following radiation.

### NANP silencing causes more DNA damage in response to radiation and impairs DNA repairs

To further explore how NANP disruption contributes to more G2/M arrest and apoptosis following radiation, we determined the level of γH2AX, a marker of DNA DSBs, whose dynamic changes reflect DNA

repair capacity and efficiency[26]. Immunofluorescence analysis (Fig. 4a, b) of γH2AX foci revealed that 1 h post radiation, nearly all cells exhibited extensive damage which was repaired over time. However, NANP deficient cells showed more γH2AX foci at 16 hours post radiation. This trend of increased γH2AX foci persisted, although to a lesser extent, at the 24-hour time point. Notably, the near-100% positivity of γH2AX at 1 hour post-radiation made it challenging to discern differences between groups at this early time point. Thus, immunoblot was performed to detect γH2AX protein levels in GSC20, GSC11 and U87. Consistent with the immunofluorescence results, higher γH2AX protein levels were observed at late time points in NANP repressed cells, and immunoblot was able to distinguish different levels of γH2AX at earlier time points of about 3 hours post-radiation when DNA damage repair is insufficient (Fig. 4c, d), indicating the possibility of more initial radiation-induced DNA damage in NANP deficient cells.

To complement these findings, we employed the alkaline comet assay, a sensitive method for assessing DNA damage. NANP suppressed cells exhibited a higher proportion of cells with elongated tails compared to control cells at 1 hour after radiation (Fig. 4e, f), validating the observation of more damage at early time point. This difference in damage level persisted as NANP silenced cells retained a higher tail moment (Fig. 4e, f) even at 24 hours when a majority of elongated comet tails were resolved in control cells as shown by the lower tail moment. This difference in damage could stem from variations in initial radiation-induced damage levels and/or DNA repair capability. To determine if more initial damage were induced in NANP silenced cells, we measured the γH2AX levels at additional time points. The results (Fig. 4g and Supplementary Fig. 9 a, b) revealed that γH2AX levels peaked at -1 h in both shCTL and shNANP cells, while it persisted higher at 1 h, 2 h, 4 h, 8 h,12 h and 24 h in NANP-deficient cells. Furthermore, in even shorter time points ( <1 h), when the time is too short to complete repairs and the initial damage was still accumulating as shown by the increase level of γH2AX, NANP-silenced cells exhibited higher γH2AX levels as early as 5 minutes post-radiation (Fig. 4h). Collectively, these data suggest that NANP silencing exacerbates initial radiation-induced DNA damage.

DSB typically activates NHEJ and/or HR repair. HR is a more accurate DNA repair mechanism and happens mainly during the S and G2 phases of the cell cycle when homologues sequences are available, while NHEJ is more error prone and happens at a much higher rate than HR since it can happen at any phase of a cell cycle[27]. With a robust reporter system[28], we were able to quantitatively compare the efficiency of HR and NHEJ. We first evaluated the system by treating cells with SCR7, a Lig4 inhibitor, to block NHEJ. With this system, we observed decreased NHEJ efficiency with no change in HR in the GSC20 treated with the Lig4 inhibitor SCR7 (Supplementary Fig. 10a, b), confirming the efficacy of this reporter system. The reporter was then used to evaluate the HR and NHEJ in NANP silenced and control cells. NANP silenced cells utilized NHEJ significantly more than control cells, while HR was significantly reduced in the same cells (Fig. 4i-k and Supplemental Fig. 10c). These data suggest NANP silenced cells favor error-prone NHEJ more over HR when repairing radiation-induced DSBs. This may partially explain the impaired repair efficacy in NANP deficient cells. Taken together, our data consistently demonstrate the

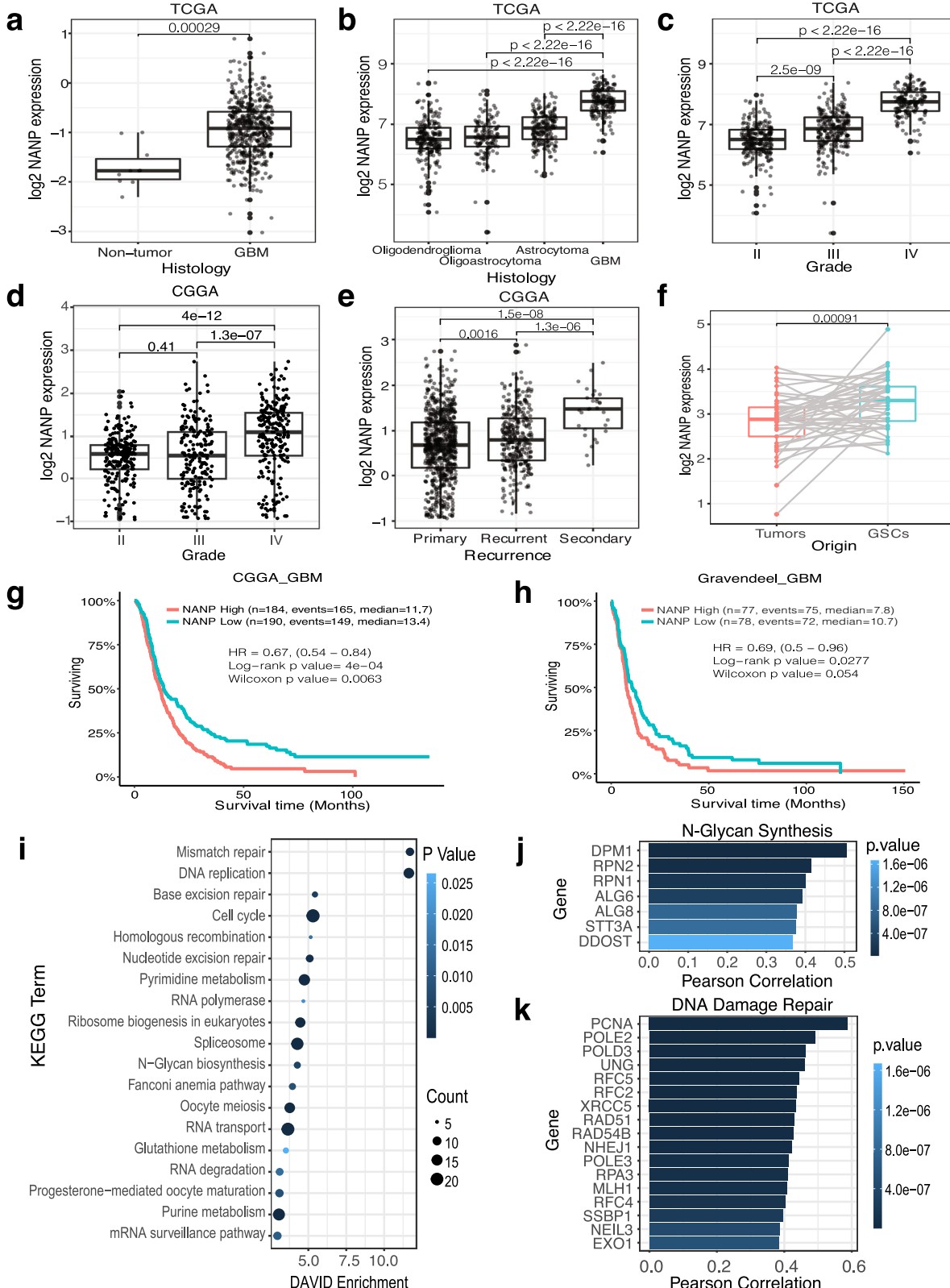

prolonged presence of more substantial DNA damage and impaired DNA repair capacity in NANP silenced cells.

## NANP regulates the mesenchymal state of GSCs

To investigate how targeting NANP affects GSC radiosensitivity, we performed whole transcriptome analysis (RNA-seq) and compared gene expression between non-targeting control and NANP silenced cells in both GSC20 and GSC11. Using a q-value < 0.05 and |log2FC| > 0.5 as cut-offs, we identified 614 genes that were significantly up-regulated and 916 genes were significantly down-regulated in NANP knock-down GSC20 compared with control cells (Fig. 5a, Supplementary Data 4). Similarly, in GSC11 we identified 579 up-regulated and 741 down-regulated genes (Supplementary Data 4). Evaluating the overlapping gene sets from these GSCs, we found 112 genes were

**Fig. 2 | NANP expression in public patient cohorts and GSCs. a** Boxplot showing the expression of *NANP* in GBM patients (*n* = 489) compared with non-tumor tissues (*n* = 10) in the TCGA dataset. Two-sided unpaired t test *p*-value is shown. **b** Boxplot showing the expression of *NANP* in GBM (*n* = 152) compared with other gliomas like oligodendroglioma (*n* = 191), oligoastrocytoma (*n* = 130), astrocytoma (*n* = 194) in TCGA dataset. Two-sided unpaired t test *p*-values are shown. **c**, **d** Boxplot showing the expression of *NANP* in gliomas by World Health Organization grade (II, III and IV) in the TCGA dataset (*n* = 226, 244, 150 for grade II, III and IV, respectively) and Chinese Glioma Genome Atlas (CGGA) dataset[61] (*n* = 291, 334, 388 for grades II, III and IV, respectively), respectively. Unpaired t-test, two-sided *p*-values are shown. **e** Boxplot showing the expression of *NANP* in primary (*n* = 651), recurrent (*n* = 333) and secondary (*n* = 30) GBM from CCGA dataset. Unpaired t-test, two-sided *p*-values are shown. **f** Boxplot showing the expression of *NANP* in GSCs and the paired bulk tumor tissue (*n* = 43). Two-sided paired t test *p*-value is shown. Box plots **a-f** show the median (center line), IQR (box: 25$^{th}$-75$^{th}$ percentiles), and whiskers extending to the minimum/maximum values within 1.5×IQR. **g**, **h** Kaplan-Meier survival analysis for overall survival between patients with high and low expression of *NANP* in CGGA and Gravendeel GBM cohorts[62], respectively. Two-sided Log-rank and Wilcoxon *p*-values are presented. **i** KEGG pathway enrichment analysis of genes highly positive correlates (Pearson correlation *r* > 0.3) with *NANP*'s expression in TCGA dataset with DAVID webtool. Bubble size corresponds to gene number enriched in term and color corresponds to term enriched *p*-value, which was assessed by two-sided Fisher's exact test with false discovery rate (FDR) adjustment for multiple comparisons. **j**, **k** Bar graph showing Pearson correlation to *NANP* expression in TCGA dataset for genes with function in N-Glycan synthesis **j** and DNA damage repair **k**. Statistical significance was evaluated by two-sided t-test.

commonly down-regulated and 67 genes were commonly up-regulated in both GSC20 and GSC11 upon NANP silencing (Supplementary Fig. 11a, b). Interestingly, functional annotation of these common differentially expressed genes using the Database for Annotation, Visualization and Integrated Discovery (DAVID) analysis revealed enrichment for genes associated with proteins containing N-linked glycosylation as well as functions in the extracellular matrix, membrane, and cell adhesion (Supplementary Fig. 11c, d). This suggested the disruption of glycosylation homeostasis upon NANP attenuation, which was consistent with the established role of NANP in glycosylation and suggests the reflection of NANP biology in the transcriptome analysis.

To identify additional involved pathways, we next performed gene set enrichment analysis (GSEA) and identified "epithelial mesenchymal transition (EMT)" as the most significant negatively regulated "hallmark" signature in both GSC20 and GSC11 upon NANP silencing (Fig. 5b). Prior transcriptome analysis of GBM have classified them into three molecular subtypes based on the putative function of the dominant gene expression signature for each class: proneural (PN), for enrichment of neural developmental genes; classical (CL), for genes related to oncogenic signaling; and mesenchymal (MES), for genes associated with mesenchymal functions[10]. The negative enrichment for the EMT pathway following NANP silencing indicated a possible role of NANP in regulating the MES subtype. This was further supported by the negative enrichment of the "Verhaak glioblastoma mesenchymal" gene set upon NANP silencing (Fig. 5c). Furthermore, we employed a classification strategy utilizing single-sample GSEA (ssGSEA) combined with equivalent distribution resampling as previously described[10]. This approach enabled us to assign three empirical classification *p*-values, facilitating the identification of significantly activated subtypes. Notably, GSC20 was classified into the mesenchymal (MES) subtype. However, following NANP knockdown, this significance diminished, and the knockdown cells exhibited a trend towards other subtypes without a significant classification (Fig. 5d), further supporting the shift from mesenchymal status following NANP silencing.

To further validate the shift towards a less mesenchymal state of GSC upon NANP attenuation, we assessed the protein levels of key markers involved in the EMT via immunoblot. We observed an increase in the epithelial marker E-cadherin and a decrease in the mesenchymal marker N-cadherin following NANP silencing (Fig. 5e), consistent with the elevated mRNA levels of *CDH1* and reduced levels of *CDH2* (Fig. 5f). Additionally, NANP silencing led to a decrease in the EMT-induced transcription factor SLUG, along with alterations in the β-catenin signaling pathway (Fig. 5e). These alterations were evident both at the protein and mRNA levels (Fig. 5e, f), which was validated by knock-out and rescue experiments (Supplementary Fig. 12a-c). Given the known impact of EMT on cell migration, we also investigated migration capacity through transwell migration assays and found decreased migration of GSC20 following NANP silencing (Fig. 5g, h). Collectively,

these findings support the conclusion that NANP positively regulates the mesenchymal state of GSCs.

## NANP regulates radiosensitivity by promoting a MES phenotype through N-linked sialylation of TNFR1, which activates the NF-κB pathway

Based on additional GSEA analysis of the RNA-seq data, we observed that NANP silencing cells were negatively enriched for genes in the NF-κB pathway (Fig. 5b and Fig. 6a). Our previous study demonstrated that NF-κB pathway activation leads to mesenchymal activation and promotes radiation resistance in GSCs[12], a conclusion supported by multiple other reports[14,15]. Considering the regulation of MES state and the NF-κB by NANP, we hypothesized that NANP regulates MES status and RT sensitivity through the regulation of NF-κB.

We first validated this finding by accessing the expression of NF-κB regulated MES genes in GSC20, GSC11 and U87 by reverse transcription-quantitative PCR (RT-qPCR). We observed a significant decrease in expression of *TNC*, *TNFAIP3/6*, *ITGB1*, and *ICAM1* (Fig. 6b). The primary mechanism for canonical NF-κB activation is mediated by tumor necrosis factor receptor (TNFR). Upon TNFα binding, TNFRs form membrane bound receptor complexes, leading to activation and phosphorylation of the IκB kinase (IKK) complex and thereby triggering ubiquitin-dependent IκBα proteasomal degradation[29]. Thus, we further validated the regulation of this key signal cascade by NANP. As expected, we observed a decrease in p-IKKα/β and an increased level of IκBα upon NANP silencing (Fig. 6c). Inhibition of NF-κB by overexpression of dominant-negative IκB (dnIκBα) led to significant downregulation of MES genes (Fig. 6d, Supplementary Fig. 13a), and the regulation of these genes by NANP was largely blunted (Fig. 6d), supporting the conclusion that the regulation of MES genes by NANP occurs via the NF-κB pathway and is dependent on IκBα. Taken together, our data suggest that NANP promotes the MES phenotype of GSCs through the activation of the NF-κB signaling pathway upstream of IκBα.

We next explored how NANP influences NF-κB signaling. Given its role in sialylation and the known effects of sialylation on EMT[30,31], we first conducted a comprehensive examination of sialylation changes through lectin screening to detect surface sialylation changes. Our analysis revealed that α2,6-linked N-sialylation detected by SNA (*Sambucus nigra agglutinin*) and α2,3-linked O-sialylation detected by MAL-II (*Maackia amurensis lectin II*) were not altered. Corresponding de-sialylation detection by PNA (*Peanut agglutinin*, β1,3-linked galactose) and WGA (*Wheat germ agglutinin*, β1,4-linked-N-Acetylglucosamine) also showed no changes, reflecting that global membrane sialylation remained unchanged (Supplementary Fig. 14). This is consistent with previous report that NANP knockout does not disrupt sialylation of cell surface glycans but decrease de novo sialic acid biosynthesis[32].

However, previous studies have shown that TNFR1 activity is significantly affected by α2,6-linked N-linked sialylation, which inhibits

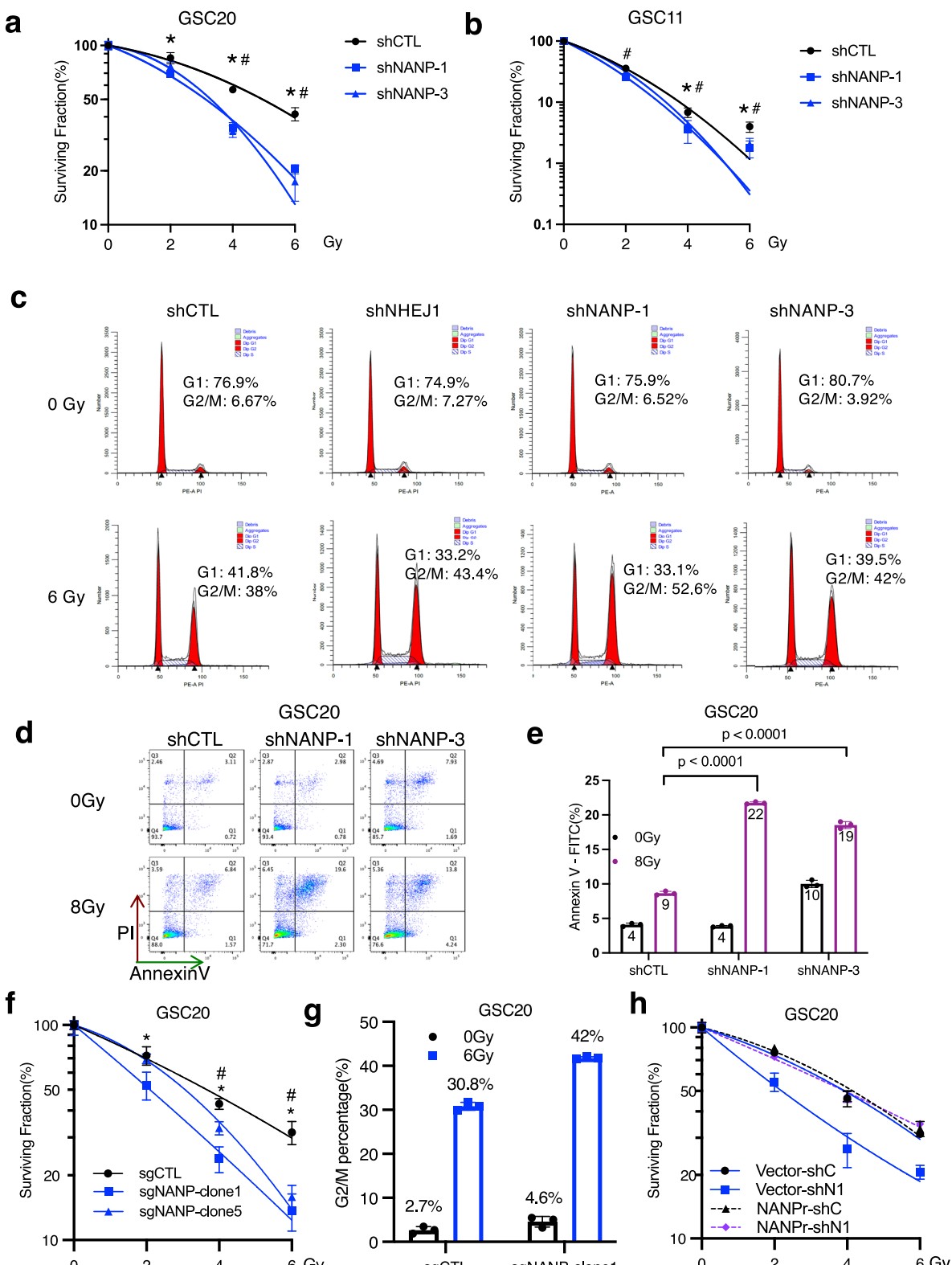

TNFR1 internalization and thereby favors signaling towards NF-κB activation[33,34]. Although we did not observe a global cell surface sialylation change after NANP silencing, we hypothesized that NANP suppression might specifically affect sialylation of TNFR1, which is upstream of IκBα, thereby inhibiting the NF-κB pathway. To test this hypothesis, we performed immunoprecipitation (IP) using SNA-agarose beads to isolate α2,6-linked sialylation and followed with

TNFR1 immunoblot. We observed decreased binding of TNFR1, but not another membrane sialoglycoprotein, podoplanin (PDPN)[35], to SNA in NANP knock down cells compared to control cells (Fig. 6e), indicating TNFR1 in NANP silenced GSC20 had reduced α2,6-sialylation. This finding was further validated by oppositely IP of endogenous TNFR1 and blot for its SNA sialylation (Supplementary Fig. 15a, b). After PNGase F digestion, we observed down shift of TNFR1 and absent of

**Fig. 3 | Targeting NANP radiosensitizes GBM. a, b** Radiation sensitivity plots showing surviving fractions of GSC20 **a** and GSC11 **b** without (shCTL) or with NANP silencing (shNANP-1 or shNANP-3). Data are means ± SD; *$p < 0.05$ shNANP-1 compared with shCTL, # $p < 0.05$ shNANP-3 compared with shCTL ($n = 4$ biological replicates, two-sided, unpaired t-test, exact $p$-values provided in the Source Data file). **c** Cell cycle analysis of GSC20 without or with NHEJ1 or NANP knockdown at 24 h after radiation (0 Gy or 6 Gy). Similar results were obtained from three independent experiments. **d, e** Apoptosis analysis of GSC20 after 48 h of radiation using Annexin V-FITC-PI staining. Data are means ± SD; $n = 3$ biological replicates; two-sided, unpaired t-test used. FITC, fluorescein isothiocyanate; PI, propidium iodide.

**f** Radiation sensitivity plots showing surviving fractions of control(sgCTL) and NANP knock-out (sgNANP) clones. Data are means ± SD; $n = 4$ biological replicates, *$p < 0.05$ sgNANP-clone1 compared with sgCTL, # $p < 0.05$ sgNANP-clone5 compared with shCTL ($n = 4$ biological replicates, two-sided, unpaired t-test). Exact $p$-values provided in the Source Data file. **g** Cell cycle analysis of GSC20 without or with NANP knock-out at 24 h after radiation (0 Gy or 6 Gy). Data are means ± SD ($n = 3$ biological replicates). **h** Radiation sensitivity plots showing surviving fractions of control (shC) or NANP knock-down(shN1) cells with shRNA-resistant NANP (NANPr) or vector expression. Data are means ± SD; $n = 4$ biological replicates, two-sided, unpaired t-test, exact $p$-values provided in the Source Data file.

SNA blotting, further demonstrated the purity of immunoprecipitated TNFR1 and specificity of SNA detection, respectively (Supplementary Fig. 15a, b).

Given that sialylation of TNFR1 affects its internalization, we performed an internalization assay as previously reported[36] to determine the impact of NANP silencing on this process. Cells were incubated with Flag- tagged TNFα for 60 min on ice, which allows TNFα binding to TNFR1, but prevents the internalization of TNFα-TNFR1 complexes. TNFR1 internalization was then examined by switching cells with bound TNFα to 37 °C for 30 min. After this interval, cells were stained with the anti-Flag antibody to assess the levels of TNFα-TNFR1 complexes remaining on the cell surface after the internalization step. We did observe significantly increased internalization of TNFR1 after TNFα binding in NANP knockdown cells evidenced by greater left-shift in density plot of surface TNFR1 (Fig. 6f and Supplementary Fig. 15c). Together, these data suggest NANP promotes TNFR1 sialylation and thereby regulates its internalization.

TNFR1 sialylation inhibits its internalization and promotes NF-κB activation[33,34]. Consistently, the basal level of NF-κB signaling was lower in NANP silenced cells as evidenced by the decrease in NF-κB target genes' expression and increase in IκBα protein level (Fig. 6b, c). Furthermore, in the presence of TNFα, NF-κB signaling induction and the nuclear p65 level was attenuated in cells where NANP was silenced (Fig. 6g, h and Supplementary Fig.13c), indicating an impaired response of TNFR to its ligand. Functionally, the increase of G2/M cell cycle arrest in response to RT by NANP suppression could be reversed by TNFα mediated NF-κB activation (Fig. 6i, Supplementary Fig. 13b), consistent with our previous study that found that TNFα induced EMT and rescued G2/M arrest after radiation in GSCs[12]. Collectively, these findings suggest that NANP regulates radiosensitivity by promoting a MES phenotype through N-linked sialylation of TNFR1 leading to activation of the NF-κB pathway (Fig. 6j).

## NANP suppression radiosensitizes GBM in vivo

To validate the radiosensitizing effect of NANP attenuation observed in vitro, we investigated its impact on tumor growth following clinically relevant fractionated radiotherapy. Immunocompromised mice received intracranial injections of control (shCTL) or NANP silenced (shNANP) GSC20 or GSC11 cells to establish orthotopic xenografts (Fig. 7a). These tumors were then treated with a fractionated regimen of 2 Gy × 5 days of RT weekly for two weeks. The results (Fig. 7b) demonstrated that RT did not significantly extend the survival of mice with control GSC20 cells implantation, which was consistent with the known radioresistant nature of GSC20. However, RT significantly extended the survival of mice receiving NANP repression GSC20 cells (log-rank $p$-value: 0.0093). Similar results were obtained in GSC11 model (Fig. 7c). Although RT alone prolonged the survival of mice implanted with control GSC11 cells, consistent with the RT-sensitive nature of GSC11, RT extended the survival of mice receiving NANP-silenced GSC11 to a greater extent (Fig. 7c). We also performed immunohistochemical staining for TNC as an indicator of NF-κB regulation in the aforementioned xenograft models at the survival endpoint (Fig. 7d). TNC staining was significantly diminished in the

shNANP group compared with the shCTL group (Fig. 7e), regardless of radiation status, validating the intrinsic suppression of the NF-κB pathway in vivo following NANP attenuation. Lastly, we explored the clinical relevance of NANP and RT by analysis of the Glioma Longitudinal Analysis consortium (GLASS) dataset[37]. NANP expression was significantly and negatively correlated with glioma patient survival of those receiving RT treatment in the GLASS cohort (Fig. 7f). In contrast, for the patients not receiving RT, there was no significant difference in survival between patients with high and low NANP expression (Fig. 7g). These findings highlight the promising radiosensitizing effect by NANP suppression in vivo, warranting future therapeutic development.

## Discussion

In this study, we aimed to identify targets to overcome resistance to RT, the most effective first-line therapy for GBM patients following surgical resection. Our clonal evolution study reveals the absence of dominant RT-resistant clones that could be analyzed for clone-specific resistance targeting. This prompted us to conduct a genome-wide CRISPR screen for RT-sensitizing targets, which identified 139 potential targets, including those well-known for their involvement in the DDR related pathways. NANP emerged as an insufficiently characterized and significant radiosensitizing target. NANP, an enzyme critical for sialic acid synthesis, plays a crucial role in modulating the sialylation of TNFR1, which in turn affects NF-κB signaling and the MES state.

The importance of the DDR pathways in radiosensitization is well-established[38]. Our study identified key NHEJ pathway genes (*PRKDC*, *NHEJ1*) as potent radio-sensitizers in the CRISPR screen, validating the effectiveness of our approach. Interestingly, NANP deficiency also impaired DNA repair, favoring the error-prone NHEJ pathway over the more accurate homologous recombination (HR) pathway. This is consistent with NANP's role in regulating TNFAIP3, a NF-κB target gene whose deletion increases the efficiency of NHEJ and decreases HR by regulating H2A turnover critical for proper DNA repair[39].

The mechanistic insights provided by our study suggest that NANP influences RT sensitivity by regulating the sialylation of TNFR1, which is crucial for NF-κB pathway activation and the maintenance of the MES state in GSCs. This is consistent with our previous findings that link NF-κB activation with increased mesenchymal differentiation and radioresistance in GSCs[12]. Additionally, our in vivo validation using an intracranial orthotopic xenograft model further supports the role of NANP in modulating radiosensitivity. Interestingly, the radiosensitivity regulated by NANP in GBM was not limited to MES subtype, as evidenced by its radiosensitizing effect on GSC11, a CL subtype model. While exhibiting dominant subtype signature expression, a GBM may retain partial expression of other subtype signatures due to intrinsic subtype plasticity—a phenomenon supported by studies highlighting the heterogeneity and dynamic transitions between GBM subtypes.[11,12] Though an inferior model of GBM compared to GSC lines, the U87 human glioblastoma cell line[40] was also included as an additional model to further confirm the generality of the phenotype. Exploring the therapeutic potential of NANP inhibitors in combination with RT for GBM treatment warrants future investigation.

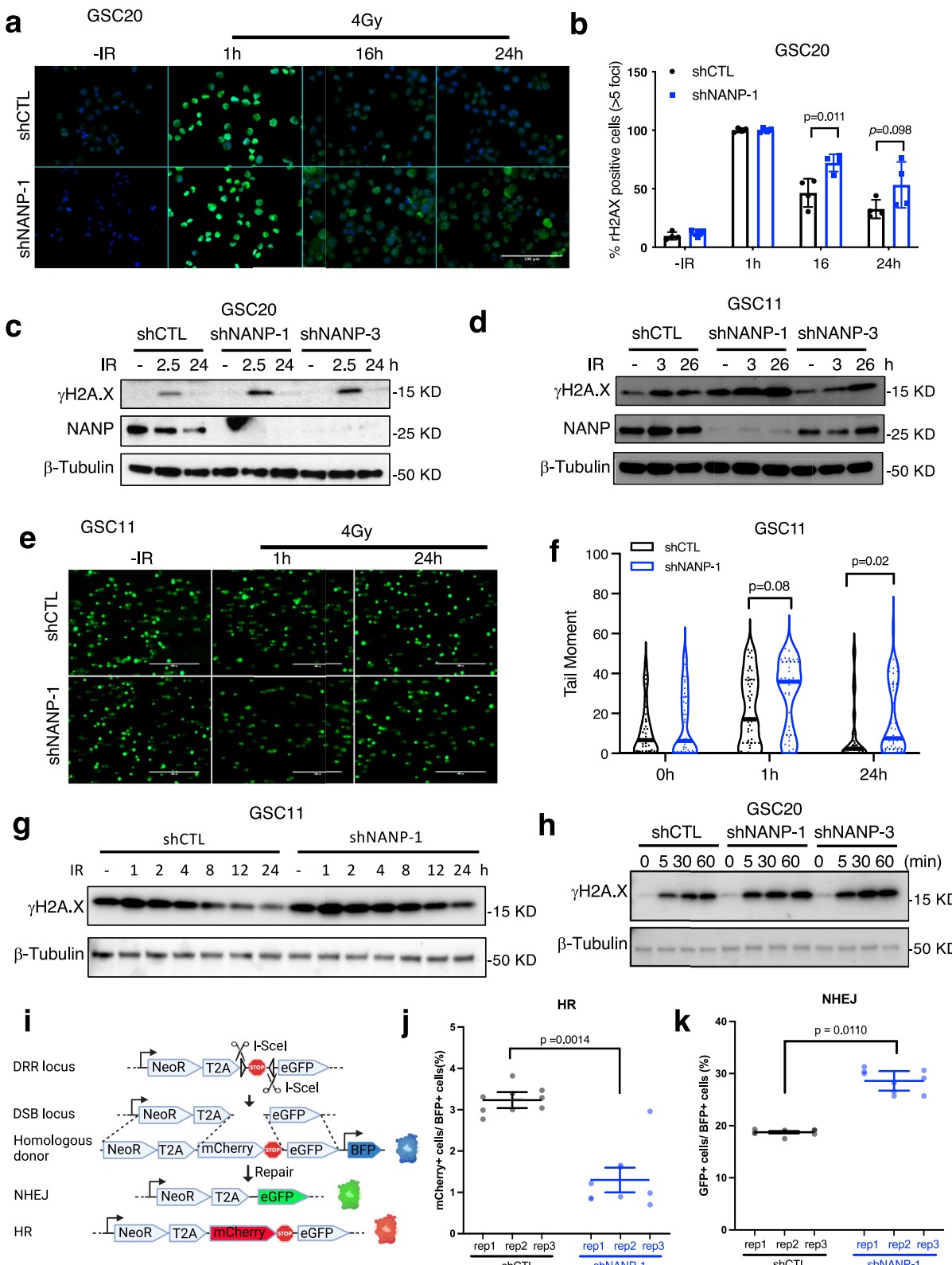

Our study demonstrates NANP disruption does not impact global cell surface protein sialylation, consistent with a prior report that showed NANP deficiency maintained normal Neu5Ac levels[32]. Rather, NANP silencing selectively impacts sialylation of TNFR1. Our study also reinforces the crucial role of sialylation in tumor progression and therapy resistance, particularly in radiotherapy[41–44].While inhibiting N-linked glycosylation with NGI-1 enhances glioma radiosensitivity by

reducing glycosylation and EGFR activation[45], this approach can affect multiple other proteins (e.g., TLR4, CD19), leading to off-target effects. Specific ablation of glycosylation components, such as STT3A, could offer a more targeted strategy[46]. Interestingly, our study demonstrated that NANP disruption selectively alters TNFR1 sialylation while maintaining normal global sialylation, making it a promising selective target for GBM radiotherapy, but future studies are required to elucidate a

**Fig. 4 | NANP disruption causes more DNA damage in response to radiation and impairs DNA repair capacity.** (**a**) Representative γH2AX foci staining of GSC20 without radiation or at indicated time points post-radiation. Green channel: the γH2AX staining; blue channel: 4′,6-diamidino-2-phenylindole (DAPI) staining for nuclei. The scale bar is 100 µm. (**b**) Quantification analysis of (**a**). Cells with >5 γH2AX foci were considered as γH2AX positive. At least 100 cells for each group were analyzed. Data are means ± SD; *n* = 4 distinct fields; two-sided, unpaired t-test used. Similar results were observed from three independent experiments. (**c** and **d**) Western blot results showing the dynamic of γH2AX levels in control (shCTL) or NANP silencing (shNANP-1 or shNANP-3) GSC20 (**c**) and GSC11(**d**) harvested at indicated time points. Loading control: β-tubulin. Three independent replicates with similar results observed. (**e**) Representative images (acquired using 4× objective lens) showing alkaline comet assay results of GSC11 without radiation or at indicated time points post-radiation. The scale bar is 1000 µm. (**f**) Quantification of (**e**) using OpenComet. Unpaired t-test *p*-values (two-tailed) are shown (shCTL: *n* = 34 cells for 0 h, *n* = 44 cells for 1 h, *n* = 52 cells for 24 h; shNANP-1: *n* = 42 cells for

0 h, *n* = 32 cells for 1 h, *n* = 44 cells for 24 h). Similar results were obtained from three independent experiments. (**g, h**) Western blot results showing the dynamic of γH2AX levels in control (shCTL) or NANP silencing GSC11 (**g**) and GSC20 (**h**) harvested at indicated time points. Loading control: β-tubulin. Three independent replicates with similar results observed. (**i-k**) Analysis of HR and NHEJ efficiency in control or NANP-deficient GSC20. (**i**) Schematic outline of the reporter system. Single-cell origin cells with one copy of DDR locus integration were used for the experiment. Transfection of I-SceI and donor plasmids to these cells induces DSB and provides homologous donor, respectively. The transfected cells are BFP positive, of which cells repaired by NHEJ will be GFP positive and those repaired by HR will be mCherry positive. Created in BioRender. Zhang, Z. (2025) https://BioRender.com/gnecstk (**j** and **k**) Quantification of HR (**j**) and NHEJ (**k**) usage by control or NANP-deficient GSC20 cells. Data are means ± SD; *n* = 3 biological replicates, each with 3 technical replicates (each shown as a point); unpaired t-test with Welch's correction; two-tailed *p*-values are presented. Similar results were obtained from three independent experiments.

broader spectrum of NANP's effects beyond sialylation on radio-sensitivity, as well as how NANP silencing affects TNFR1 sialyation while sparing global sialylation. Moreover, our study did not rule out the possibility of NANP having additional substrates, such as fructose 1,6-bisphosphate, a concept proposed by the first publication describing NANP[47] which may also impact the DDR.

In conclusion, our findings identify NANP as a promising radio-sensitizing target in GBM. The dependency of NANP's effects on TNFR1 sialylation and the MES shift underscores the complex interplay between sialylation, signaling pathways, and cellular phenotypes in modulating radiosensitivity. Further research is crucial to fully understand the mechanisms by which NANP regulates these processes and to translate this knowledge into improved therapy for patients with GBM.

## Methods

All experimental procedures involving animals and human-derived samples in this study were conducted in strict compliance with the relevant ethical regulations, including the National Institutes of Health (NIH) Guide for the Care and Use of Laboratory Animals and the Declaration of Helsinki. The animal study protocol was approved by the Institutional Animal Care and Use Committee (IACUC) of NYU Langone Health.

### Cell culture, DNA construction, lentiviral transduction and cell line generation

GSCs were developed in our laboratory as described previously[12]. GSC cells were maintained in neural basal media (NBM) containing DMEM/F12 medium (Corning, 10-090-CV), supplemented with B27 (Gibco, 17504-044), epidermal growth factor (EGF) (20 ng/ml, Sigma, E9644), basic fibroblast growth factor (bFGF) (20 ng/ml, Sigma, F0291) and 1% antibiotic/antimycotic supplements (Corning, 30-004-CI). For cell passaging, GSCs were dissociated into single cells with Accutase (Sigma, A6964) once large neurospheres formed. U87 (ATCC, HTB-14) and 293 T (ATCC, CRL-3216) cells were grown in DMEM with 10% fetal bovine serum (FBS) (Sigma, F0926) and 1% antibiotic/antimycotic supplements (Corning, 30-004-CI). Lentiviral plasmids for targeting NANP were ordered from Sigma (TRCN0000050489 for shNANP-1 or shN1 in short; TRCN0000050488 for shNANP-3 or shN3 in short; TRCN0000050491 for shNANP-4; TRCN0000050492 for shNANP-5 and TRCN0000275632 for shNHEJ1), and empty vector was used as control. Lentiviral plasmid for expressing dominant-negative IκB (dnIκBα) was constructed by inserting the dnIκBα expressing DNA amplified from pBABE-Zeo-IkB-SR[48] into a lentiviral vector pHAGE-BSD with blasticidin selection marker. Lentiviral plasmid for expressing shRNA-resistant NANP was cloned by overlapping PCR to introduce wobble codons at the shN1 target region using cDNA as template, and

pHAGE-BSD was used as vector. Lenti-CRIPSRv2-Blast (Addgene, # 83480) was used to generate guide RNAs (sgNANP: ACTCTCATCGA-CACGGCCG; sgCTL: GCGAATACGCCCACGCGAT) for knock-out. For virus production, 293 T cells were transfected with lentiviral core plasmid and lentiviral packaging plasmids psPAX2 (Addgene, #12260) and pMD2.G (Addgene, #12259) at a ratio of 4:3:1. The media was changed to NBM medium 12 h after transfection in the virus was used to transduce GSCs in order to avoid introducing of FBS to GSCs. Viral supernatant was collected 72 h after transfection and filtered with a 0.45 µm filter (Millipore) and then used for cell transduction at a multiplicity of infection (MOI) of 0.3 in the presence of polybrene (Sigma, TR-1003-G). 48 hours after transduction, cells were washed and selected with 2 µg/ml puromycin or 5 µg/ml blasticidin for 10 days. For CRISPR-mediated knockout, the bulk transduced cells survive the selection were first expanded to get enough cells and then used to validate the overall gene knockout by western blot. The bulk cells that showing target gene protein decreased were then seeded into multiple 96-well plates for single-cell clone isolation by limited dilution approach. Single-cell clones were then isolated and expanded to obtain enough cells for western blot validation and other experiments, such make freezing stocks, colongenic assays, cell cycle analysis etc.

### Clonal evolution of RT resistance in GSCs

The clonal evolution study was performed as previous described[19]. Briefly, 1 million GSC20 cells were barcoded using the CAPTURE barcoding library[19] at a MOI < 0.3. The founder barcoded cells were then expanded and divided into a parallel control group and 5 RT-treated replicates, each containing $10^8$ cells (>100× barcode diversity). The cells from the RT-treated group were subjected to fractioned radiation treatments (2 Gy × 10), which were delivered daily for two weeks (5 days/week). The control cells were cultured in parallel. After the last dose of radiation, the cells were continued in culture for another 35–40 days to allow for regrowth of the population. Cells (>100× barcode diversity) were harvested for barcode deconvolution. TruSeq-style barcode amplicon libraries were amplified and subjected to deep sequencing using an Illumina sequencer. The sequencing reads were mapped and counted as previously described[19]. The GFOLD (generalized fold change)[49] algorithm was employed to produce reliable statistics based on the posterior distribution of log2 fold change, which was visualized using the pheatmap R package.

### CRISPR screen

The Brunello genome-wide CRISPR-Cas9 library[50] was used in lentiviral pooled format to transduce GSCs. The experiment was performed in duplicate. For each screen replicate, cells were transduced at ~300-fold representation of the library (at 30% infection efficiency). 2 days after

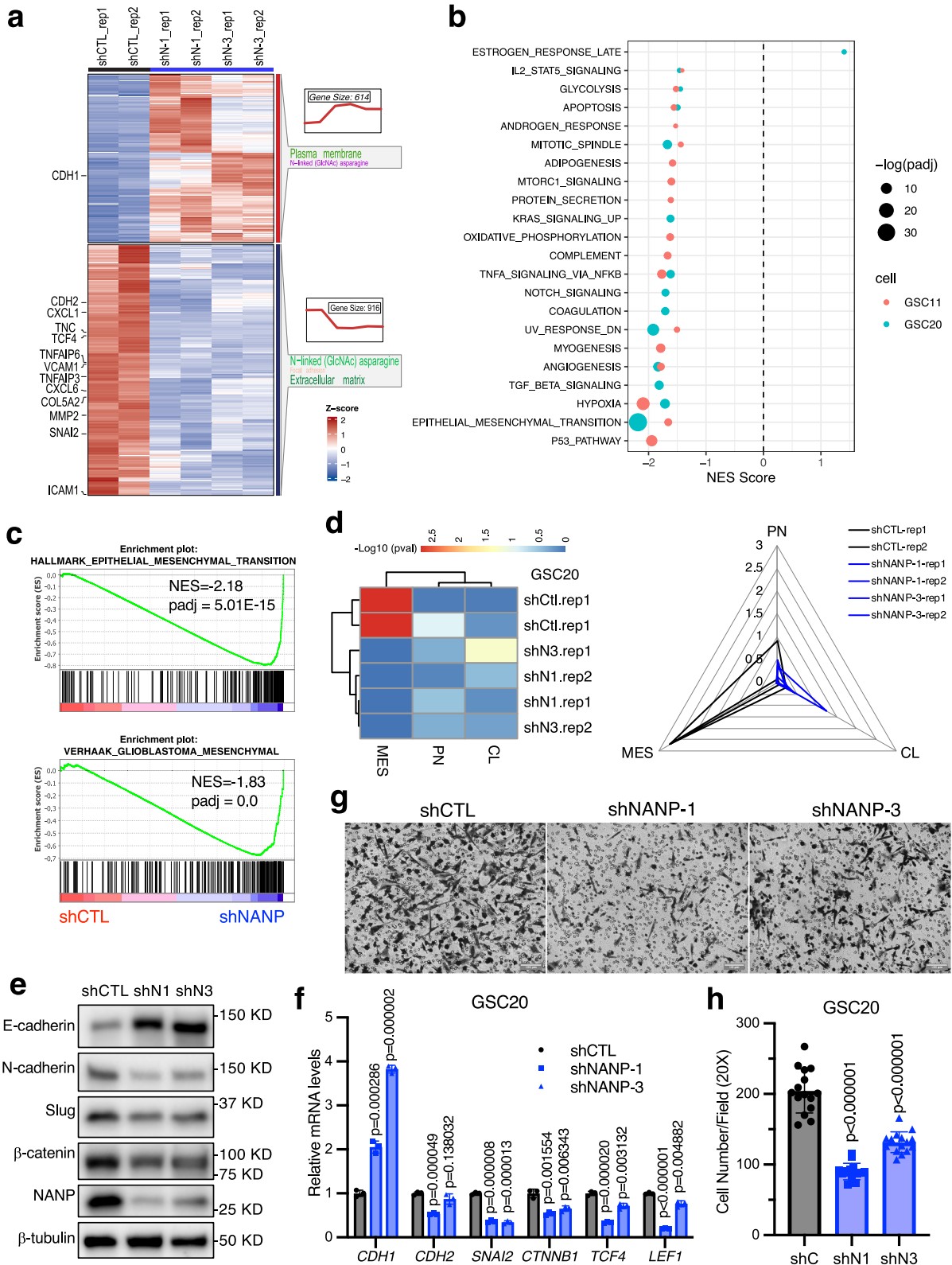

transduction, puromycin was added (2 μg/ml) for 7 days. The cells were then passaged to maintain 300-fold representation and cultured for an additional 25 days (>8 cell doublings). Genomic DNA was extracted using Quick-DNA Midiprep Plus Kit (Zymo Research, D4075). TruSeq-style amplicon PCR was performed to amplify single guide RNAs (sgRNAs). The amount of genomic DNA used for PCR was calculated to maintain 300-fold coverage over the library. The resulting

amplicon PCR products were pooled for each experimental replicate and purified using AMPure XP Bead (Beckman Coulter, A63880), followed by concentration quantification using KAPA Library Quantification Kits (Roche, 07960140001) and sequencing using a HiSeq 4000 (Illumina). The resulting reads as FASTQ files were analyzed using MAGeCK[51] to generate sgRNA- and gene- level count and ranking tables, which were inputted for further analysis and visualization using

**Fig. 5 | NANP regulates the mesenchymal state of GSCs.** (**a**) Heatmap showing differential expression genes ( | log2FC | > 0.5, $q < 0.05$) between control and NANP silenced GSC20 visualized by GlusterGVis. Genes involved in EMT and NF-κB pathways are shown on the left side of the figure. The top enriched GoTerm or KEGG pathway are shown on the right side with size corresponding to -log10 ($p$ value) for that term. (**b**) Bubble plot showing significantly enriched hallmarks by GSEA analysis for GSC20 (blue bubble) and GSC11(red bubble). Bubble size corresponding to -log10 (padj). (**c**) GSEA enrichment plot for EMT signature and Verhaak glioblastoma mesenchymal signature gene set in NANP repression versus control GSC20. (**d**) Subtype classification analysis of the three subtype classes of control and NANP silenced GSC20. Heatmap showing the significance of each subtype indicated by -log10 ($p$-value). In the radar plot showing the connection of GSCs

(control or NANP silenced) towards each subtype; black lines for control cells, blue lines are for NANP silenced cells. (**e**) Western blot showing expression of E-cadherin, N-cadherin, Slug, β-catenin in control or NANP silenced GSC20. The β-tubulin was used as loading control. Three independent replicates with similar results observed. (**f**) The mRNA expression levels of genes involved in EMT in control or NANP silenced GSC20 quantified by qPCR. Data are means ± SD; two-sided, unpaired t-test used ($n = 3$ biological replicates). (**g**) Representative images showing the migration ability comparison between NANP silenced versus control GSC20 cells. The images were acquired using a 20× objective lens. The scale bar is 100 μm. (**h**) Quantification analysis of (**g**). Data are means ± SD; two-sided, unpaired t-test used ($n = 15$ fields examined over three independent experiments).

MAGeCKFlute[52]. The top hits were subjected to Gene Ontology (GO) and Reactome pathway network analysis using Enrichr-KG[53], and the result was visualized by Cytoscape 3.

### Cell cycle analysis
Cells were collected by centrifugation and digested to single cells. Cells were washed with phosphate buffered saline (PBS) once then fixed with ice-cold 66% ethanol-PBS solution overnight at 4 °C. Then cells were stained using the Propidium Iodide Flow Cytometry Kit (Abcam, ab139418) according to the manufacturer's instructions. Briefly, cells were stained with 200 μl 1× propidium iodide (PI) plus RNase staining solution at 37 °C in the dark for 30 min then analyzed using a LSRII HTS flow cytometer (BD Biosciences). Data analysis was performed using FlowJo (Becton Dickinson) or ModFit LT (Verity Software House).

### Apoptosis analysis
Cells were collected by centrifugation and resuspend into single cell suspension followed by two rounds of PBS washing. Apoptosis assays were performed using the FITC Annexin V apoptosis Detection Kit I (BD pharmingen, 556547). Briefly, cells were resuspended in 1× binding buffer and stained with FITC AnnexinV and PI at room temperature in the dark for 15 min and then analyzed using a LSRII HTS flow cytometer (BD Biosciences). Data analysis was performed using FlowJo (Becton Dickinson).

### Clonogenic assay and cell viability assay
Exponentially growing U87 cells were seeded in 6-well plates (3000 cells/well) overnight before treatment with radiation. Radiation was performed using a CellRad X-Ray Irradiator (Precision X-Ray) at a dose rate of approximately 1 Gy/min. U87 cells were allowed to grow for 10–14 days after radiation. Media was changed every three days. Cells were then fixed with 4% paraformaldehyde (PFA) and stained with Crystal violet. For cell viability assays, 1000 cells with four replicates were seeded to 96-well plates and then given corresponding radiation treatment. Cells were allowed to grow in the incubator for 10–14 days with media changed and replaced every three days. Cell viability was measured by CellTiter-Glo (Promega, G7571). The clonogenic assay for GSCs were performed as previously described[54]. Survival fraction and survival curves were fitted with a Linear Quadratic model using Prism 9 (GraphPad).

### Immunofluorescence γH2AX staining
GSCs were seeded on Poly-D-Lysine/Laminin coated 12 mm round cover glasses (Corning BioCoat, 354087) in 24-well plates. The following day GSC cells were attached and treated with radiation. After treatment at the indicated timepoints cells were fixed with 4% PFA followed by permeabilization and blocking in blocking solution [1% bovine serum albumin (BSA), 0.1% Triton X-100 in PBS] at room temperature for 30 min. Cells were then stained with γH2AX antibody (Millipore Sigma, 05-636, 1:1 K in blocking solution) at 4 °C overnight. After washing with PBS-T (PBS containing 0.1% Tween-20) for three

times, cells were stained with goat anti-Mouse IgG (H + L) Highly Cross-Adsorbed Secondary Antibody, Alexa Fluor™ Plus 488 (Thermo, A32723, 1:2 K in blocking solution) at room temperature for 1 h. After washing with PBST for three times. Stained cells were mount with ProLong™ Glass Antifade Mountant with NucBlue™ Stain (Invitrogen, P36983). Images were acquired by Zeiss Axio Observer microscope (Zeiss) and analyzed by ImageJ. Nuclei with more than 5 foci were considered as γH2AX positive.

### Comet assay
GSC cells were harvested and suspended as single cells at the indicated time after radiation. Alkaline comet assay was performed with CometAssay kit (Trevigen, 4250-050-K) according to the manufacturer's instructions. Briefly, freshly harvested cells were mixed with low-melting point agarose and spread on specially treated CometSlides. Cells on the slides were then lysed at 4 °C for 1 h and then in alkaline unwinding solution for 30 min at room temperature in the dark. The slides were then subjected to horizontal electrophoresis at 21 V for 30 min under alkaline conditions, followed by neutralization and SYBR green staining. During electrophoresis, damaged DNA migrates away from the nucleus forming a comet-like tail, while undamaged DNA remains in the head region. Images were acquired by the EVOS® FL Auto Imaging System (Life Technologies) and tail moment were analyzed by OpenComet[55].

### Immunoblot
Cells were harvested at indicated time points after treatment and washed with PBS once, followed by lysing with RIPA buffer (CST, 9806). Protein concentration was measured by Pierce™ BCA Protein Assay Kit (Thermo, 23225). After SDS-PAGE and transfer, membranes were blocked with 5% BSA in TBST (TBS buffer containing 0.1% Tween-20) and probed with antibodies against phospho-H2AX (Ser139) (MilliporeSigma, 05-636), NANP (SantaCruz, sc-374637), E-cadherin (CST, 3195), N-cadherin (CST, 13116), Slug (CST, 9585), β-catenin(CST, 8480), pospho-IKKα/β (Ser176/180) (CST, 2697), IKKα (CST, 11930), IKKβ (CST, 8943), phospho-p65 (CST, 3033), p65 (CST, 8242), IκBα (CST, 1814), PDPN (CST, 9047). Tubulin probed by anti-tubulin (Biolegend, 801202) was used as loading control. Membranes were incubated with appropriate horse radish peroxidase (HRP)-linked anti-rabbit IgG (CST, 7074) or anti-mouse IgG (CST, 7076) secondary antibodies and protein bands were detected by Amersham ECL Prime Western Blotting Detection Reagent (Cytiva, RPN2236) or Clarity™ Western ECL Substrate (Biorad, 1705060) with ChemiDoc MP Imaging System (Biorad).

### HR/NHEJ reporter assay
The lentiviral pLCN DSB Repair Reporter (DRR) was previously described[28]. GSC20 single clone was picked after transduced with viral particle of the DDR at MOI < 0.1. Gene silencing was done on this single clone by lentivirus transduction. For DNA repair reporter analysis, one million cells containing the integrated reporter were transfected with 4 μg pDonor HR plasmid (Addgene, #98896, pCAGGS DRR mCherry

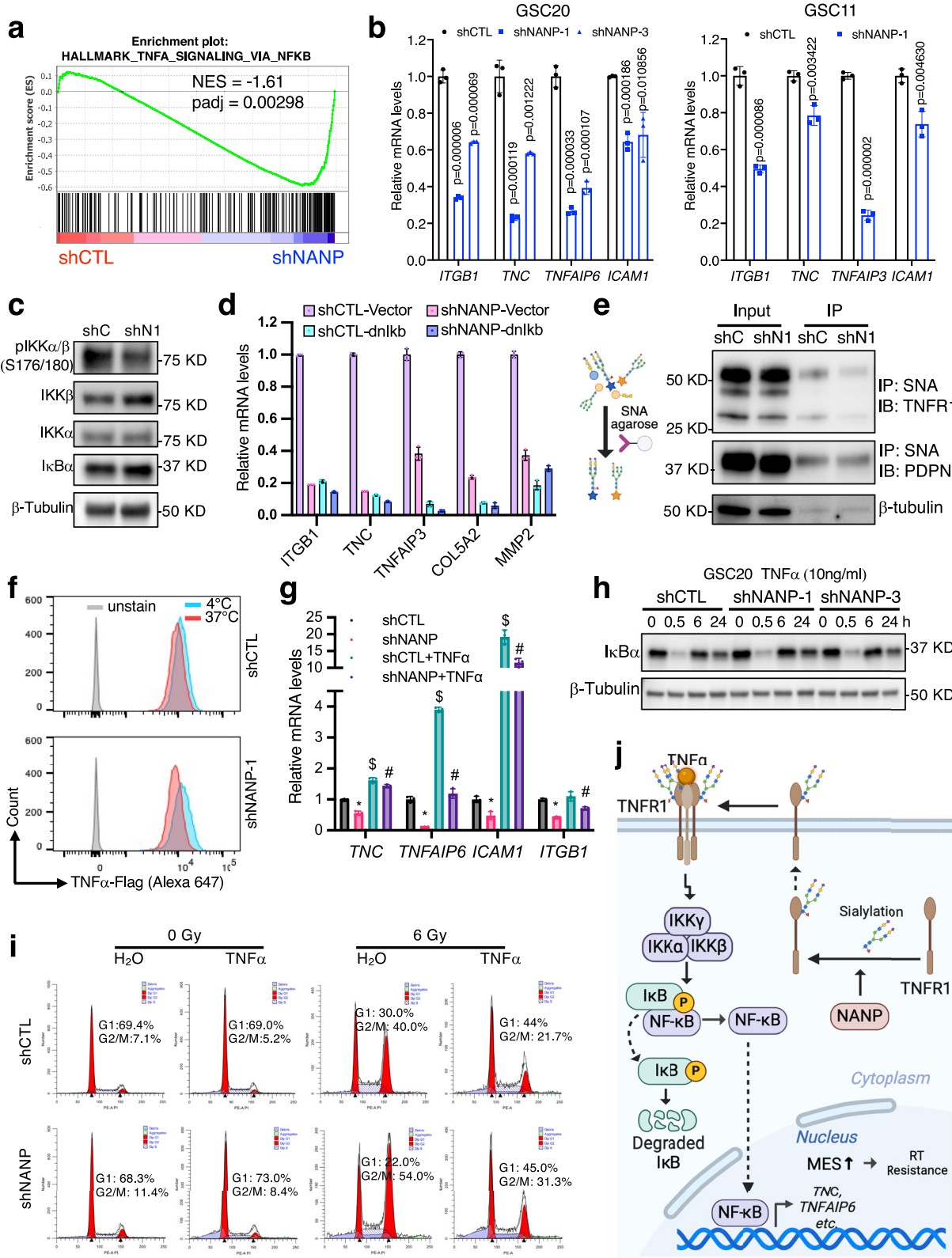

Donor EF1a BFP) and 2.5 μg pCBASceI plasmid (Addgene, #26477) using Lipofectamine 3000 (Thermo Fisher, L3000015). After 72 h of transfection, cells were digested to single cells and resuspended in 1× PBS and directly subjected to flow cytometry analysis (BD, LSRII HTS). Green fluorescent protein (GFP) and mCherry positive cells were gated on blue fluorescent protein (BFP) positive cells to access the NHEJ and HR, respectively.

## RNA-seq and data analysis

RNA-seq was performed similarly as previously described[19]. Briefly, total RNA was extracted using RNeasy Plus Mini Kit (Qiagen, 74134), and RNA-seq libraries were prepared using the Illumina TruSeq Stranded mRNA kit according to the manufacturer's instructions. Paired-end reads were sequenced on an Illumina sequencer. Reads were trimmed for adapters and quality using Trimmomatic version

**Fig. 6 | NANP regulates radiosensitivity by promoting a mesenchymal (MES) phenotype through N-linked sialylation of TNFR1, which activates the NF-κB pathway. a** GSEA enrichment plot for "TNFα signaling via NF-κB" hallmark gene set in RNA-seq analysis of shNANP vs shCTL GSC20 cells. **b** The mRNA expression levels of NF-κB regulated genes in control and NANP repression GSC20 and GSC11. Data are means ± SD; $n = 3$ biological replicates; unpaired t-test p-value(two-tailed) are shown. **c** Western blot showing protein levels of pIKKα/β (S176/180), IKKα/β, IκBα in control (shC) and NANP repression (shN1) GSC20. Loading control: β-tubulin. Three independent replicates with similar results observed. **d** The mRNA expression levels of NF-κB regulated genes in control and NANP silenced GSC20 with IκB inhibition using dominant-negative IκB (dnIκB) overexpression or control (vector). Data are means ± SD, $n = 3$ biological replicates. **e** TNFR1 expressed by NANP knockdown GSC20 had reduced α2,6 sialylation compared with control cells. GSC20 whole cell lysate were precipitated with agarose-conjugated SNA and blotted for TNFR1, PDPN and β-tubulin. Three independent replicates with similar results observed. The diagram Created in BioRender. Zhang, Z. (2025) https://BioRender.com/gnecstk. **f** TNFR1 internalization analysis by flow cytometry

detection of cell surface TNFα binding before (4 °C) and after (37 °C) induction of internalization in control (shCTL) and NANP knock-down (shNANP-1) cells. Quantification results were presented as Supplementary Fig. 15c. Similar results were observed from two independent experiments. **g** The effect of TNFα treatment (10 ng/ml, 24 h) on the mRNA levels of NF-κB regulated genes levels in control and NANP repression GSC20. Data are means ± SD. $^*p < 0.05$ shCTL with vehicle ($H_2O$) vs. shNANP with vehicle; $^\$p < 0.05$ shCTL with TNFα vs. shCTL with vehicle; $^\#p < 0.05$ shNANP with TNFα vs. shNANP with vehicle. ($n = 3$ biological replicates, two-sided, unpaired t-test used). **h** Western blot showing IκBα protein levels in NANP silenced versus control GSC20 without or with TNFα (10 ng/ml) treatment at indicated timepoints. Loading control: β-tubulin. Two independent replicates with similar results observed. **i** Cell cycle analysis of NANP knockdown versus control GSC20 with or without TNFα (10 ng/ml, 24 h) stimulation and with or without radiation. Three independent experiments with similar results observed. **j** Schematic depicting the proposed model of NANP regulating TNFR1 sialylation and subsequent NF-κB signaling, MES status and RT sensitivity of GBM. Created in BioRender. Zhang, Z. (2025) https://BioRender.com/gnecstk.

0.36 followed by aligning to the human reference genome (hg19) using STAR version 2.5 with standard settings. The featureCounts program was employed to generate a genes-to-samples counts matrix. Differentially expressed genes were analyzed using DESeq2. Gene set enrichment analysis (GSEA) was performed using GSEA. Transcriptome-based GSC subtyping was performed using ssgsea.GBM.classification R package[10]. RNA-seq data has been submitted to the Gene Expression Omnibus (GEO) repository (GSE274135).

### Transwell cell migration assay
Transwells with 8.0 μM pore polycarbonate membrane Insert (Corning, 3422) was precoated with poly-l-ornithine (20 μg/mL) (Sigma, P3655) and laminin (5 μg/mL) (Sigma, L2020). GSC20 carrying control shRNA or NANP shRNA ($3 \times 10^4$ cells/insert) were seeded into the upper chamber in three replicates. NBM with 10% FBS were added to the bottom chamber as a chemo attractant. Cells were allowed to migrate towards the bottom in a humidified incubator at 37 °C with 5% $CO_2$ for 48 h. Then the cells on the membrane were fixed with 4% PFA and stained with crystal violet. Images were taken by EVOS® FL Auto Imaging System (Life Technologies) with 8-12 fields per membrane. Quantification of cells was performed using ImageJ.

### Reverse transcription quantitative -polymerase chain reaction (RT-qPCR)
Total RNA was isolated from cells after treatment with RNeasy Plus Mini Kit (Qiagen, 74134) and then reverse transcribed into cDNA using Maxima H Minus First Strand cDNA Synthesis Kit (Thermo, K1652). RT-qPCR was performed on Applied Biosystems Vii7 cycler with PowerUp™ SYBR™ Green Master Mix (Thermo, A25742). Primers used are listed in Supplemental Data 5.

### Cell fractionation
The cells were fractionated with NE-PER Nuclear and Cytoplasmic Extraction Reagents (Thermo Scientific, 78833) according to manufacturer's instruction. Immunoblots were performed following nuclear extraction. CTCF (CST, 2899) was used as the indicator of nuclear fraction and Tubulin (BioRad, 12004166) was used as the indicator of cytoplasmic fraction.

### Lectin screening
For detecting cell surface sialylation, $5 \times 10^5$ GSC20 cells were suspended as single cells then washed with staining buffer (1× PBS, 1 mM $CaCl_2$, 1 mM $MgCl_2$, 1% BSA) for three times. Cells were subsequently incubated with biotinylated *Sambucus nigra agglutinin* (SNA, 4 μg/ml, Vector Labs, B-1305-2), biotinylated *Maackia amurensis Lectin II* (MAL-II, 4 μg/ml, Vector Labs, B-1265-1), biotinylated peanut agglutinin (PNA,

4 μg/ml, Vector Labs, B-1075-5) or biotinylated *Wheat germ agglutinin* (WGA, 4 μg/ml, Vector Labs, B-1025-5) in staining buffer at 4 °C degree with rotation for 1 h. Cells stained with blank staining buffer without lectin were used as negative controls. The cells were then washed with cold staining buffer and then incubated with Streptavidin-conjugated Alexa Fluor 488 (2 μg/ml, Invitrogen, S11223) at 4 °C degree with rotation for 1 h. After washing with cold staining buffer three times, the fluorescence of the labelled cells was detected by flow cytometry (LSRII HTS, BD).

### SNA-pull down assay
For lectin-IP assay, exponentially growing shCTL and shNANP GSC20 were collected and washed once with PBS. Pellets were then lysed with RIPA buffer (CST, 9806) supplemented with Halt Protease and Phosphatase inhibitor cocktail (Thermo, 78442). 1000 μg of whole cell lysate was incubated with 50 μl of SNA-conjugated agarose beads (VectorLab, AL-1303-2), which were prewashed with RIPA buffer, overnight at 4 °C with rotation. The α2-6-sialylated proteins bound to the beads were then precipitated by centrifugation, followed by washing with RIPA buffer for three times to remove nonspecific binding. The sialylated proteins were then eluted from the beads using 50 μl 1× Laemmli sample buffer by boiling at 98 °C for 5 min. Parallelly, 100 μg of whole cell lysate prepared using 1× Laemmli sample buffer was used as input. Proteins were resolved by SDS-PAGE and immunoblotted for TNFR1 (CST, 3736) and ß-tubulin (Biolegend, 801202).

### TNFR1 glycosylation analysis
Exponentially growing shCTL and shNANP GSC20 were collected and washed once with PBS. Pellets were then lysed with RIPA buffer (CST, 9806) supplemented with Halt Protease and Phosphatase inhibitor cocktail (Thermo, 78442) for 1 h on ice. After centrifugation at 16,000 rpm at 4 °C, the supernatants were collected. 5% of the supernatants were prepared using 1× Laemmli sample buffer as input while remaining supernatant were incubated with TNFR1 primary antibody conjugated agarose beads (SantaCruz, sc-8436 AC) at 4 °C overnight. The beads were washed three times with extensive wash buffer (50 mM Tris-HCl, pH 7.4, 500 mM NaCl, 1 mM EDTA and 1% Triton X-100) at 4 °C and one more time with modified RIPA buffer (50 mM Tris-HCl, pH 7.4, 150 mM NaCl, 1 mM EDTA. Washed beads were divided equally to two aliquots, one aliquot was used to directly elute the protein with 2× SDS sampling buffer (80 mM Tris-HCl, pH 6.8, 2% SDS, 10% glycerol) by boiling at 98 °C for 5 min. The other aliquot was digested with PNGase F (NEB, P0704) according to manufacturer's instructions. Briefly, proteins were first eluted with 1× denaturing buffer (0.4% SDS, 40 mM DTT) at 100 °C for 10 min. After cooling to room temperature, SDS was

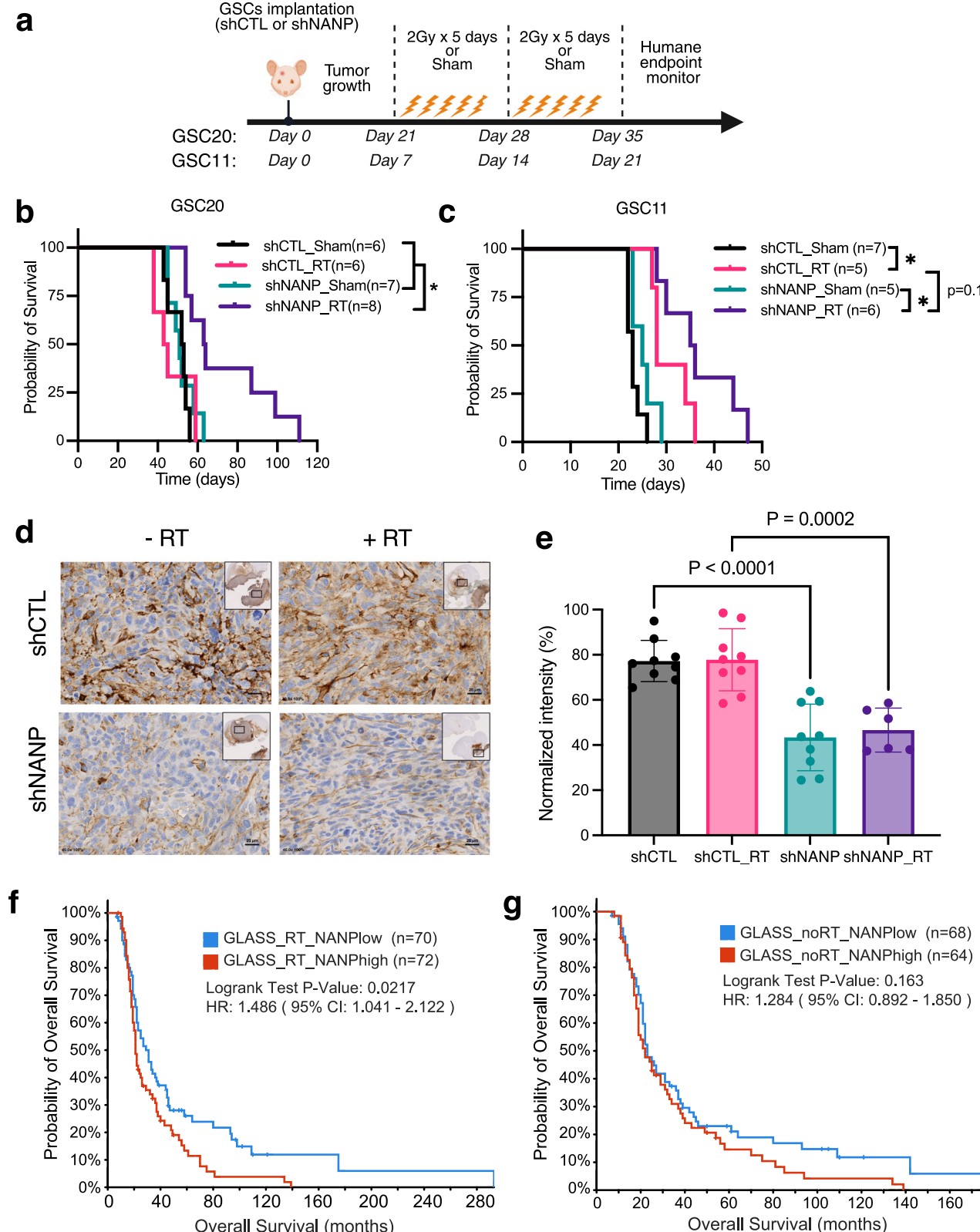

**Fig. 7 | NANP suppression radiosensitizes GBM in vivo. a** Schematic outline of the experimental design for the in vivo experiment. Created in BioRender. Zhang, Z. (2025) https://BioRender.com/gnecstk. **b, c** Kaplan-Meier survival analysis of mice treated as indicated. *$p < 0.05$, logrank (**b** shCTL_shSham vs. shNANP_RT *p*-value = 0.0068, shNANP_shSham vs. shNANP_RT *p*-value = 0.0243; **c** shCTL_shSham vs. shCTL_RT *p*-value = 0.0008, shNANP_shSham vs. shNANP_RT *p*-value = 0.0025). **d** Immunohistochemical staining for TNC in GBM xenografts treated as indicated

are shown. **e** Quantification of fraction of TNC positive area and intensity to total area in the indicated group is shown. Three distinct fields for each mouse (*n* = 3 mice for each group, except *n* = 2 mice for shNANP_RT group) were analyzed. Data are means ± SD. Adjusted *p*-values by Tukey's multiple comparisons test are presented. **f, g** Kaplan-Meier survival analysis of glioma patient from GLASS cohort with **f** or without **g** RT. Median expression of *NANP* was used as the cutoff for high versus low.

neutralized with 10% Nonidet P-40 (NP-40). Glycans were digested at 37 °C for 4 h using 1 U PNGase F per 10 μg proteins. For immunoblot analysis, the membranes were blocked with use carbon free blocking solution (VectorLab, SP-5040-125) at room temperature for 1 h. Then the membranes were incubated with fluorescein (FITC) conjugated Sambucus Nigra Lectin (Thermo, L32479) and Rhodamine conjugated Ricinus Communis Agglutinin I (RCA I, RCA120) (VectorLab, RL-1082-1) at same time at room temperature for 3 hr in PBS-T (PBS containing 0.05% Tween-20). After washed three times wash of PBS-T, bands were detected with ChemiDoc MP Imaging System (BioRad). Silver staining was performed with Pierce™ Silver Stain Kit (Thermo, #24612) according to manufacturer's instructions.

### TNFR1 internalization assay

Exponentially growing shCTL and shNANP GSC20 were treated with 200 ng/ml TNFα with Flag tag (Enzo Life Sciences, ALX-522–008) at 4 °C for 1 h. Unbound TNF-Flag was washed off with ice-cold PBS twice. Cells were then incubated with prewarmed 37 °C serum-free medium for 30 min to induce the internalization of TNF-TNFR1 complexes. Control cells were maintained at 4 °C. The internalization step was then terminated by placing cells back on ice. To measure the levels of surface TNF-TNFR1 complexes, cells were collected and washed with ice cold PBS once and then stained with anti-Flag-Alexa 647 (1:200, Invitrogen, 701629RP647) for 30 min on ice. Propidium Iodide (PI) (Sigma Aldrich, P4170, 2 μg/ml) was used to exclude dead cells. Data were acquired on a BD FACSymphony™ A5 flow cytometer and analyzed by the FlowJo (Becton Dickinson). The relative amount of internalization was calculated by subtracting MFI values obtained after the 37 °C internalization step from MFI values obtained at 4 °C prior to the internalization step.

### Animal experiments

All animal studies were performed in accordance with protocols approved by IACUC of NYU Langone Health. Mice were housed under specific pathogen-free (SPF) conditions in a temperature-controlled room maintained at 22 ± 2 °C with a relative humidity of 50 ± 10%. A 12-hour light/12-hour dark cycle was implemented (lights on at 07:00 AM, lights off at 07:00 PM), and mice had ad libitum access to standard laboratory chow and filtered water. 6-8 weeks old BALB/c Nude (nu/nu) were used in this study. All mice were randomly assigned to appropriate treatment groups. For intracranial orthotopic implantation, $5 \times 10^5$ cells of dissociated GSCs were stereotactically injected into the left striatum of nude mice using a guide-screw technique[56]. Female mice were used for GSC20 experiments, and equal numbers of male and female mice were initially used for GSC11 experiments (a small number were lost during guide-screw placement/implantation). Moreover, GSC20 and GSC11 were derived from male and female patients, respectively, therefore sex was not considered a biological variable. The GSCs were transduced by pLenti-PGK-V5-LUC Neo (Addgene, #21471) viral particles to express luciferase for monitoring tumor growth using an in vivo imaging system (IVIS) bioluminescence imaging system. Fractionated radiation (2 Gy for 5 consecutive days) to the whole brains of mice with a small field biological irradiator, the SARRP Irradiator from Xstrahl with image-guided radiation capability was performed and the treatment was delivered with the small 1.5 mm cylinder. Moribund state, inability to rise or ambulate were used as humane endpoints, and spontaneous death was also recorded for mouse survival. The endpoints were monitored daily by trained staff working in the animal facility, who would attach identification tags to mice that met the endpoint criteria and promptly notify the laboratory technicians for subsequent handling, in strict accordance with the guidelines of the IACUC at NYU Langone Health. Brain tissues were harvested at the endpoint for immunohistochemistry (IHC) analysis.

### Immunohistochemistry (IHC)

For IHC, brains were fixed in 4% PFA followed by paraffin-embedding. Five-micron thick sections were immune-stained using a Leica BondRX automated stainer according to the manufacturer's manual. Briefly, tissues underwent deparaffinization online, followed by epitope retrieval for 20 min at 100 °C with Leica Biosystems ER2 solution (pH9, AR9640) and endogenous peroxidase activity blocking with $H_2O_2$ (provided in the Leica BOND Polymer Refine Detection System, DS9800). Sections were then incubated with primary antibodies against TNC (CST, 93029) at 1:600 for 30 min at room temperature. Primary antibodies were detected with anti-rabbit HRP-conjugated polymer and 3,3′-diaminobenzidine substrate that are provided in the Leica BOND Polymer Refine Detection System. Following counter-staining with hematoxylin, slides were scanned at 40× on a Hamamatzu Nanozoomer (2.0HT) and the image files uploaded to the NYUGSoM's OMERO Plus image data management system (Glencoe Software). TNC positive areas and intensity were quantified with IHC image analysis Toolbox[57].

### Database analysis

The *NANP* expression level in GBM patients and expression of correlated genes and the relation to patient survival data from clinical cohorts were obtained from GlioVis[58]. The analysis of *NANP* expression in RT-related patient survival from the GLASS cohort was performed using cBioPortal[59]. Overrepresenting pathway analysis of NANP correlated genes was performed using DAVID[60].

### Statistics

Unless otherwise noted, all tests were carried out in three biological replicates. All results were presented as mean ± SD unless otherwise stated with statistical significance determined by tests as indicated in figure legends. The analysis was performed with Prism 9 software (GraphPad) or R (version 4.1) unless otherwise noted in the methods.

### Reporting summary

Further information on research design is available in the Nature Portfolio Reporting Summary linked to this article.

## Data availability

All RNA-seq data were deposited in the NCBI GEO repository under accession GSE274135. Raw data for the genome-wide CRISPR screen and clonal evolution analysis have been deposited in the NCBI Sequence Read Archive (SRA) repository under accession number PRJNA1432116 (https://www.ncbi.nlm.nih.gov/sra). Other data that support the findings of this research are available within the paper and its Supplementary Information. Source data are provided with this paper.

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

## Acknowledgements

We thank Dr. Michael Pacold at NYU Langone Health for helpful discussions. We thank the NYU Langone Health's Cytometry and Cell Sorting Laboratory for cytometry support; Experimental Pathology Research Laboratory for support on IHC experiment; Microscopy Laboratory for microscope imaging support and Preclinical Imaging Laboratory for small animal imaging support. These cores are all partially supported by the Cancer Center Support Grant P30CA016087 at NYU Langone's Laura and Isaac Perlmutter Cancer Center. The Preclinical Imaging Laboratory is also partially supported by NIBIB Biomedical Technology Resource Center Grant NIH P41 EB017183. This work is supported by R01CA282756 (Y.D., Z.-Y.Z., and E.P.S.). This project was also supported by a grant from the NYU Technology Opportunities & Ventures Therapeutics Alliances Fund (Y.D., Z.-Y.Z.).

## Author contributions

E.P.S., Z.-Y.Z., and Y.D. conceived and designed the study; Y.D. and Z.-Y.Z. designed, performed, and analyzed most experiments; Z.-Y.Z., R.E., and M.G. performed in vivo study. G.M. and A.J. provided help with the experiments during revision. A.M., J.K. provided critical suggestions; R.E. did administration and management. Y.D., Z.-Y.Z., and E.P.S. wrote the manuscript. All authors have reviewed and revised the manuscript.

## Competing interests

The authors declare no competing interests.
