## [Transparent Peer Review file · Nature Communications]

NANP Targeting Radiosensitizes Glioblastoma through TNFR1 Sialylation-Driven Mesenchymal Shift

Corresponding Author: Professor Erik Sulman

Version 0:

Reviewer comments:

Reviewer #1

(Remarks to the Author)

Ding et al present data supporting a model whereby NANP supports the mesenchymal phenotype of GBM cells, including a more radioresistant phenotype, and hence targeting NANP can serve to radiosensitize GBM. The CRISPR screen is well-described with clear rationale for the pipeline leading to hits to focus on with strong bioinformatic support for the ultimate focus on NANP. Overall, the data provided are supportive for the relevance of NANP to the response to radiotherapy as well as for the model proposed linking NANP to the NF- κ B pathway. There are some minor concerns to be addressed to further support the impact of this work on the brain tumor field.

1. The authors cannot claim anything regarding mitotic failure as this was not evaluated, "Together, these findings suggest that NANP silencing resulted in increased mitotic failure and subsequent apoptosis following radiation." This conclusion should be revised to reflect what the data demonstrate.
2. The authors report decreased migration upon NANP depletion but also report an increase in G2/M arrested cells and an increase in apoptosis following NANP depletion. As arrested cells and dying cells will not migrate, how did the authors control for this when evaluating a decrease in migration? That is, is there a direct decrease in migration or just more cells dying/arrested in the cell cycle that aren't migrating?
3. As U87 is well known in the field to no longer represent GBM, it is unclear why these data were included and what they will contribute to the field versus the data with the GSC lines. The limitation of this model should at least be acknowledged in the Discussion.
4. The authors should go through the work and make sure all conclusions are supported by more than one shRNA (e.g., clonogenic assay in 6G/H is only with shNANP-1).

Additional

1. The intro highlights the TCGA/Verhaak subtypes but more recent single cell studies have further refined the extent of plasticity and heterogeneity and should be noted (i.e. Neftel et al., Cell, 2019).
2. Main text states fractionated regimen was 2 Gy x 5 days but Methods states 2.5 Gy x 4 days.
3. Methods only mention U87 for the clonogenic assay but this was done in the other cells as well so details should be added.

Reviewer #2

(Remarks to the Author)

Ding et al present a potentially very interesting hypothesis that NANP downregulation may sensitize glioblastoma for radiation therapy. Additionally, they claim that this effect is mechanistically explained by a role of NANP in TNFR1 specific sialylation resulting in an effect on NF κ B signaling.

Many of the experiments that are shown and discussed indeed indicate a link between NANP and RT sensitization. Causality could/should be increased by studying the effect in NANP knockout GSC20 cells and its NANP corrected lines.

Mechanistically, in my view no evidence is presented for a defect in sialylation of TNFR1. The SNA agarose IP is not a quantitative test that can be used to assess percentage of sialylation of an individual protein. As alternative, IP of TNFR1 (with proof of purity) should be performed followed by western blotting with a sialic acid specific lectin (showing a decrease in staining) as well as a lectin against the underlying galactose residue (showing elevated staining). Therefore, the claim of a TNFR1 specific sialylation defect as mechanistic explanation for increased radiosensitization is in my view not supported by experimental evidence.

Minor points:

Summary

NANP (N-acetylneuraminase-9-phosphatase): NANP (N-acetylneuraminase 9-phosphatase) and consistent naming throughout the manuscript

Many abbreviations, some unexplained (TNFR1), complicates readability

Introduction:

Despite the potential benefits that each of these advances (has)

TCGA: explain abbreviation

Results:

to identify dominant resistant clones

To identify novel (targets?), we

a global surface sialylation change

Figure 5D: why does shNANP-3 results in much higher residual NANP expression in GSC11 as compared to GSC20?

Reviewer #3

(Remarks to the Author)

In this manuscript, the authors identify a novel radiation sensitizing target, NANP, utilizing a genome-wide CRISPR screen. They demonstrate that silencing of NANP reduces NF- κ B signaling, potentially through impaired TNFR1 sialylation, leading to a decline in mesenchymal markers. Given that mesenchymal phenotype has been associated with radioresistance, this suggests a mechanism underlying the radiosensitization seen with NANP silencing. Their results correlate well with patient tumor expression profiles and endpoints, and they validate NANP suppression mediated radiosensitization in one in vivo xenograft model. They also demonstrate effects of NANP on modulation of DNA repair, though these findings are less convincing. Overall, most of the results and methods are presented clearly and the manuscript is well-written. A novel radiation sensitizing target, if fully validated, would be highly significant given the great clinical need for addressing radioresistance in GBM. However, there are some points to be addressed as detailed below.

Major points:

1. The data in Fig. 5 are not fully consistent with the authors' conclusion that NANP silenced cells display a DSB repair defect, favoring NHEJ over HR. Specifically, Fig. 5C and 5F demonstrate increased DNA damage as measured by gamma-H2AX and alkaline comet assay at 1 and 2.5 hours. This cannot be explained by the decrease in HR observed in the reporter assay in Fig. 5H, given that HR requires 4-6+ hours to complete and almost all repair at these early timepoints is NHEJ mediated. This could be interpreted as either a decrease in NHEJ activity (inconsistent with Fig. 5I) or potentially an increase in DNA damage, which has not been ruled out. Of note, Fig. 5B shows no difference in gH2AX foci at 1 hour; however 100% of cells are positive making the cut off used incapable of detecting a difference. In addition, it is not clear why the alkaline comet assay, rather than the neutral comet assay was chosen given the focus on DSB repair.

2. There is not a clear mechanistic explanation for how NANP silencing specifically effects TNFR1 (while not impacting global sialylation), making it important to validate this finding more fully. Given the extensive use of NANP silencing, a rescue experiment with NANP re-expression should be performed. Experiments investigating TNFR1 localization upon NANP suppression and determining the dependence of radiosensitization on the sialyltransferase ST6Gal-I would significantly improve the validity of the findings, again given that it is not clear mechanistically how NANP is having this specific effect.

Minor points:

1. RT doses shown in Figs. 1 and 8 are different than those described in the methods.

2. In Supplemental Fig. 3 and 4, NANP protein levels via western blotting should be shown upon shRNA knockdown in GSC20 and GSC11.

3. In Fig. 5D, there is minimal NANP knockdown with shNANP3 despite similar effects on RT sensitivity (Fig. 4b). This should be addressed. As per major point 2 above, the specificity of NANP knockdown should be confirmed via rescue experiment.

4. The conclusion in lines 187-189 is confusing and doesn't seem to agree with data in Supplemental Fig. 6.

5. In Fig. 7E, are there other sialylation target proteins that could be probed as controls to show the specificity of the decrease in TNFR1 pull-down?

6. Most of the experiments are performed in GSC20 with some reproduced in GSC11. It is noted that GSC20 resembles the mesenchymal subtype. What about GSC11? Will these results be limited to mesenchymal type tumors? Overall, the findings would benefit from reproduction in a wider array of models.

Reviewer #4

(Remarks to the Author)

In this manuscript, the authors conducted a CRISPR knockout screen to identify radiosensitizing targets in glioblastoma (GBM). They further investigated the molecular mechanism of one target gene, NANP. Overall, this is an interesting study; however, I have the following comments:

Major comments:

The authors used CRISPR knockout screening to identify radiosensitizing targets but validated NANP using only shRNAs. At least some key experiments should be confirmed using NANP sgRNA knockout.

For the in vivo radiotherapy (RT) treatment experiments, at least two different glioma stem cell (GSC) lines should be used. The levels of P65 protein, its phosphorylation, and its nuclear localization should be assessed in the NANP knockdown cells.

Minor comments:

Scale bars should be included in Figures 5A and 6G.

The cutoff for high versus low NANP expression should be indicated in Figures 8E and 8F.

Can NANP be targeted by any small molecule inhibitors?

Version 1:

Reviewer comments:

Reviewer #1

(Remarks to the Author)

The authors have adequately addressed all prior concerns from this reviewer.

Reviewer #2

(Remarks to the Author)

The authors have expanded the number of experiments to confirm and respond to the questions of the reviewers, which is highly appreciated. Knockout of NANP in the GSC20 cell line was performed and 2 clones were selected (1 and 5) with clear NANP protein decrease on western blot (supplemental Figure 8a). Experimental details are very limited on their generation. Clone 1 was used to confirm the effect on RT, while the rescue by wildtype transfection was done again on the shRNA line. It is unclear why clone 5 was shown but not tested for effect on RT and why the wildtype rescue was not performed on the knockout clones (as a more convincing model).

The causal link of the claimed effect of TNFR1 specific desialylation on RT has been addressed by an IP of TNFR1, followed by SNA lectin staining and the results section has been adapted on page 8. In panel 1 of supplemental Figure 15a, I don't see convincing evidence of a single band that was stained with SNA in lanes 3-4, or even a band with lower intensity in the shN1 lane (4) as compared to the shC lane (3). For RCA (second panel), 2 bands are specifically stained that disappear after PNGaseF, which is as expected. This may indicate that a significant portion of TNFR1 is already desialylated in normal conditions, but at least should increase in intensity upon NANP knock-down. What is worrying is that the band of IB TNFR1 (panel 3) shows increased migration in the gel (lower MW) upon PNGaseF treatment (lanes 3-4 versus lanes 5-6; as expected because entire N-glycans are removed), but the silver stained bands (panel 4) in lanes 3-6 move at exactly the same height, suggesting the same MW. In my view, you would also expect a lower MW in the silver stained bands upon PNGaseF treatment. Another question is why these experiments were done in the shRNA model, which is more complicated than the now available KO model.

All in all, the added data (especially knockouts) increase evidence of the effect of NANP on RT, but in my view these data are not convincing to support a role of NANP in TNFR1-specific sialylation and this claim should in my view be omitted from title and paper. Possibly, NANP has additional functions or otherwise results in the clear effect of NANP on RT, which may be discussed.

Reviewer #3

(Remarks to the Author)

The authors have addressed the comments thoroughly, and overall the additional data supports their conclusions. There is

one minor concern with the new data shown in Supplementary Fig. 12b, as two of the three MES/NF- κ B genes show minimal restoration of expression with NANP rescue. Analyzing a larger panel of genes (such as that done initially in Fig. 5f and 6g) would better indicate whether NANP rescue restores this expression phenotype as the authors conclude. There also seem to be some incorrect figure references in lines 215-226.

Reviewer #4

(Remarks to the Author)

The authors have addressed all my concern!

Version 2:

Reviewer comments:

Reviewer #2

(Remarks to the Author)

I would like to thank the authors for clarifying the use of different gradient gels to explain the difference in PNGaseF treated mobility of TNFR1. The use of the additional SNA-Cy5 reagent produces much cleared bands with convincing bands of IP TNFR1 stained with SNA-Cy5. This indicates that TNFR1 is partly sialylated in this model in the control situation. If I am correct, lanes 3-4 should show a lower intensity SNA staining for the shN versus shC in Figure S15b. This may seem the case. In combination with the lower capturing (IP) of TNFR1 after SNA staining, I feel that the combined results at least suggest a slight reduction in sialylation of TNFR1 and can agree with the conclusions drawn in the paper.

POINT-BY-POINT RESPONSE

We sincerely thank all four reviewers for their careful evaluation and constructive comments which we have addressed point by point as detailed below. We have also revised the manuscript to comply with editorial formatting requirements (e.g., abstract word count reduced to 149). All changes are yellow highlighted, and new figures are underlined when referred in rebuttal letter. In addition, please note changes to figures and supplementary figures as follows:

Renamed:

Figure 1 -> Supplementary Figure 1

Figure 2 -> Figure 1

Figure 3 -> Figure 2

Figure 4 -> Figure 3

Figure 5 -> Figure 4

Figure 6 -> Figure 5

Figure 7 -> Figure 6

Figure 8 -> Figure 7

Supplementary Figure 1 -> Supplementary Figure 2

Supplementary Figure 2 -> Supplementary Figure 3

Supplementary Figure 3 -> Supplementary Figure 4

Supplementary Figure 4 -> Supplementary Figure 5

Supplementary Figure 5 -> Supplementary Figure 6

Supplementary Figure 6 -> Supplementary Figure 7

Supplementary Figure 7 -> Supplementary Figure 10

Supplementary Figure 8 -> Supplementary Figure 11

Supplementary Figure 9 -> Supplementary Figure 13

Supplementary Figure 10 -> Supplementary Figure 14

Added:

Figure 3: f, g and h

Figure 4: g and h

Figure 6: f

Figure 7: c

Supplementary Figure 4: b

Supplementary Figure 5: b

Supplementary Figure 8: a, b and c

Supplementary Figure 9: a and b

Supplementary Figure 12: a, b and c

Supplementary Figure 13: c

Supplementary Figure 15: a and b

Updated:

Figure 2: d

Figure 5: g and h

Figure 6: e

Figure 7: a

All the figures referred below in the responses are new figure names, the new data referred were underlined.

REVIEWER COMMENTS

Reviewer #1 (Remarks to the Author):

Ding et al present data supporting a model whereby NANP supports the mesenchymal phenotype of GBM cells, including a more radioresistant phenotype, and hence targeting NANP can serve to radiosensitize GBM. The CRISPR screen is well-described with clear rationale for the pipeline leading to hits to focus on with strong bioinformatic support for the ultimate focus on NANP. Overall, the data provided are supportive for the relevance of NANP to the response to radiotherapy as well as for the model proposed linking NANP to the NF- κ B pathway. There are some minor concerns to be addressed to further support the impact of this work on the brain tumor field.

1. The authors cannot claim anything regarding mitotic failure as this was not evaluated, "Together, these findings suggest that NANP silencing resulted in increased mitotic failure and subsequent apoptosis following radiation." This conclusion should be revised to reflect what the data demonstrate.

RESPONSE: We appreciate your suggestion. As suggested by the reviewer, we revised the text to remove references to mitotic failure and instead conclude: "Together, these findings suggest that NANP silencing resulted in increased G2/M cell cycle arrest and subsequent apoptosis following radiation. " This aligns with our observed G2/M arrest and apoptosis induction.

2. The authors report decreased migration upon NANP depletion but also report an increase in G2/M arrested cells and an increase in apoptosis following NANP depletion. As arrested cells and dying cells will not migrate, how did the authors control for this when evaluating a decrease in migration? That is, is there a direct decrease in migration or just more cells dying/arrested in the cell cycle that aren't migrating?

RESPONSE: We appreciate your feedback. Here, we would like to clarify that: (a) decreased migration upon NANP depletion was due to transcriptional subtype shift, which does not result in G2/M arrest or apoptosis without radiation; (b) G2/M arrest and apoptosis in NANP silencing cells were induced after radiation. The increased G2/M arrest and apoptosis of NANP silencing cells only happened when cells were treated with radiation. The cell migration was evaluated without radiation, thus not affected by radiation-induced arrest/apoptosis.

3. As U87 is well known in the field to no longer represent GBM, it is unclear why these data were included and what they will contribute to the field versus the data with the GSC lines. The limitation of this model should at least be acknowledged in the Discussion.

RESPONSE: We appreciate your feedback. We acknowledge U87's limitations due to its mis-identification. Although its patient of origin is unknown, U87 is of CNS origin and is likely to be a bona fide human glioblastoma cell line based on transcriptional similarity analysis (PMID: 27582061). Thus, we included it to validate some of the findings in addition to the results obtained from GSCs. As suggested, we have highlighted the model's utility while noting caveats in discussion (line 409).

4. The authors should go through the work and make sure all conclusions are supported by more than one shRNA (e.g., clonogenic assay in 6G/H is only with shNANP-1).

RESPONSE: Thank you for the suggestion. We understand the reviewer's concern about off-target effects of shRNA. To address this concern, we included more than one shRNA for this mentioned experiment (**Fig.5 g-h** in the revised manuscript). Moreover, we performed knockout and rescue experiments for key conclusions (**Fig. 3 f-h; Supplementary Fig. 8 a-c; Supplementary Fig. 12 a-c**).

Additional

1. The intro highlights the TCGA/Verhaak subtypes but more recent single cell studies have further refined the extent of plasticity and heterogeneity and should be noted (i.e. Neftel et al., Cell, 2019).

RESPONSE: Thank you for the suggestion. We have now revised the intro to include the suggested statement (see line 57-58 of the revised MS).

2. Main text states fractionated regimen was 2 Gy x 5 days but Methods states 2.5 Gy x 4 days.

RESPONSE: Our apologies for the typo while writing using an old method template. The correct regimen is 2 Gy x 5 days (total 10 Gy), which was revised (line 679).

3. Methods only mention U87 for the clonogenic assay but this was done in the other cells as well so details should be added.

RESPONSE: Thank you for the suggestion. The details of clonogenic assay for GSCs were added to the methods section.

Reviewer #2 (Remarks to the Author):

Ding et al present a potentially very interesting hypothesis that NANP downregulation may sensitize glioblastoma for radiation therapy. Additionally, they claim that this effect is mechanistically explained by a role of NANP in TNFR1 specific sialylation resulting in an effect on NFkB signaling.

Many of the experiments that are shown and discussed indeed indicate a link between NANP and RT sensitization. Causality could/should be increased by studying the effect in NANP knockout GSC20 cells and its NANP corrected lines.

Mechanistically, in my view no evidence is presented for a defect in sialylation of TNFR1. The SNA agarose IP is not a quantitative test that can be used to assess percentage of sialylation of an individual protein. As alternative, IP of TNFR1 (with proof of purity) should be performed followed by western blotting with a sialic acid specific lectin (showing a decrease in staining) as well as a lectin against the underlying galactose residue (showing elevated staining). Therefore, the claim of a TNFR1 specific sialylation defect as mechanistic explanation for increased radiosensitization is in my view not supported by experimental evidence.

RESPONSE: We sincerely appreciate the reviewer's insightful comments and valuable suggestions, which have significantly strengthened the rigor of our study.

Regarding the causality of NANP in radiosensitization, we agree that knockout and rescue experiments are critical to validate this relationship. We have generated NANP knockout GSC20 cells using CRISPR-Cas9 and established NANP-corrected lines by reintroducing wild-type NANP into these knockout cells. The results show that NANP knockout recapitulates the radiosensitization phenotype observed in NANP-silenced cells (**Fig. 4f-g; Supplementary Fig. 8a-b, 12a**), while NANP re-expression reverses this effect (**Fig. 4h; Supplementary Fig.8c, 12b-c**), directly confirming that NANP deficiency is causally linked to enhanced radiosensitivity in GBM.

To address the mechanistic evidence for TNFR1-specific sialylation defects, we have performed endogenous TNFR1 IP with strict validation of IP purity (**Supplemental Fig. 15a**) by silver staining, confirmed the specific enrichment of TNFR1. As suggested, we further analyzed these IP samples using both sialic acid-specific lectin (SNA) and galactose residue-specific lectin (RCA). In addition, the specificity

of lectin binding was validated by PNGase F digestion. The data confirmed a decrease in SNA staining (Supplemental Fig. 15a), indicating reduced sialylation of TNFR1. However, we did not observe a concurrent increase in RCA staining of TNFR1 in NANP-silenced cells (Supplemental Fig. 15a). This may be attributed to the complex intracellular glycosylation modification network in our experimental system. It is possible that other glycosyltransferases or deglycosylases are activated when sialic acid is absent, modifying the exposed galactose residues and thus not necessarily resulted in an increase of RCA staining. Despite the lack of increased RCA staining, the decrease in SNA staining strongly supports the reduction in TNFR1 sialylation upon NANP silencing.

Moreover, since sialylation regulates TNFR1 internalization and downstream NF- κ B signaling—a key mediator of radiation resistance—we performed TNFR1 internalization assays. The results (Fig. 6f and Supplementary Fig. 15b) show increased TNFR1 internalization in NANP-silenced cells (evidenced by a left shift in surface TNFR1 density plots), which is associated with impaired NF- κ B activation (Fig. 6a-d, 6h; Supplemental Fig. 13a, 13c). Reduced NF- κ B activation weakens the cell's ability to resist radiation-induced damage (PMID: 29232556; PMID:23993863), thereby enhancing radiosensitivity. These findings link NANP-mediated TNFR1 sialylation to altered receptor trafficking, signaling, and ultimately, radiation response.

Together, these additional experiments address the reviewer's concerns and provide robust evidence for the proposed mechanism.

Minor points:

Summary

NANP (N-acetylneuraminase-9-phosphatase): NANP (N-acetylneuraminase 9-phosphatase) and consistent naming throughout the manuscript
Many abbreviations, some unexplained (TNFR1), complicates readability

RESPONSE: Thank you. We have corrected these abbreviations and carefully checked other abbreviations throughout the manuscript.

Introduction:

Despite the potential benefits that each of these advances (has)
TCGA: explain abbreviation

RESPONSE: Thank you. We have amended as suggested.

Results:

to identify dominant resistant clones
To identify novel (targets?), we
a global surface sialylation change

RESPONSE: Thank you. We have corrected all these grammatical errors.

Figure 5D: why does shNANP-3 results in much higher residual NANP expression in GSC11 as compared to GSC20?

RESPONSE: We appreciate the reviewer's careful observation. We have rechecked the knockdown efficiency of all NANP-targeting shRNAs using both qPCR and immunoblot (Supplementary Figs. 4a-b; Supplementary Fig. 5a-b), and the results from both methods are consistent: shNANP-3 indeed results in higher residual NANP expression in GSC11 than in GSC20.

Such cell line-specific differences in shRNA efficiency are commonly observed in RNAi experiments, and multiple factors may contribute to this phenomenon. These include cell-type-specific epigenetic modifications (e.g., mRNA methylation), cell-type-specific binding of regulatory proteins (e.g., RNA-

binding proteins that can mask the shRNA target site), divergence in RNAi machinery activity, and/or sequence polymorphisms in the target region.

While investigating the precise mechanism underlying this cell line-specific difference would be of interest, it lies beyond the primary focus of the current study, which centers on the functional role of NANP in radiosensitization. However, we have confirmed that even with the higher residual NANP expression in GSC11, shNANP-3 still induces the expected radiosensitization phenotype in this cell line (**Fig. 3b, 4d; Supplementary 5; 9b**), supporting the functional relevance of NANP knockdown in both models.

Reviewer #3 (Remarks to the Author):

In this manuscript, the authors identify a novel radiation sensitizing target, NANP, utilizing a genome-wide CRISPR screen. They demonstrate that silencing of NANP reduces NF- κ B signaling, potentially through impaired TNFR1 sialylation, leading to a decline in mesenchymal markers. Given that mesenchymal phenotype has been associated with radioresistance, this suggests a mechanism underlying the radiosensitization seen with NANP silencing. Their results correlate well with patient tumor expression profiles and endpoints, and they validate NANP suppression mediated radiosensitization in one in vivo xenograft model. They also demonstrate effects of NANP on modulation of DNA repair, though these findings are less convincing. Overall, most of the results and methods are presented clearly and the manuscript is well-written. A novel radiation sensitizing target, if fully validated, would be highly significant given the great clinical need for addressing radioresistance in GBM. However, there are some points to be addressed as detailed below.

Major points:

1. The data in Fig. 5 are not fully consistent with the authors' conclusion that NANP silenced cells display a DSB repair defect, favoring NHEJ over HR. Specifically, Fig. 5C and 5F demonstrate increased DNA damage as measured by gamma-H2AX and alkaline comet assay at 1 and 2.5 hours. This cannot be explained by the decrease in HR observed in the reporter assay in Fig. 5H, given that HR requires 4-6+ hours to complete and almost all repair at these early timepoints is NHEJ mediated. This could be interpreted as either a decrease in NHEJ activity (inconsistent with Fig. 5I) or potentially an increase in DNA damage, which has not been ruled out. Of note, Fig. 5B shows no difference in gH2AX foci at 1 hour; however 100% of cells are positive making the cut off used incapable of detecting a difference. In addition, it is not clear why the alkaline comet assay, rather than the neutral comet assay was chosen given the focus on DSB repair.

RESPONSE: We appreciate your astute observation regarding the interpretation of DNA damage repair phenotypes. We share your interpretation of the mentioned data and would like to clarify that we did not conclude a sole cause-effect relationship between the HR-NHEJ shift and the DNA damage outcome. In fact, we agree with the reviewer that the damage outcome could stem from variations in initial radiation-induced damage levels and/or DNA repair capability.

As noted, the 100% positivity of γ H2AX foci at 1 hour limits the ability to detect differences at this time point. To address this, we performed further time-course γ H2AX immunoblot experiments in GSC20 and GSC11 (using two shRNAs per cell line) to assess damage at 1 hour, a time point when repair is minimally complete. The results (**Fig. 4g** and **Supplementary Fig. 9**) showed increased DNA damage in NANP-silenced cells as early as 1 hour post-radiation. In addition, we observed that DNA damage accumulated shortly after radiation in GSCs, peaked at \sim 1 hour (**Fig. 4g-h**). Notably, NANP-silenced cells exhibited increased damage as early as 5 minutes post-radiation (**Fig. 4h**), a time that is too short to complete repairs and the initial damage was still accumulating as shown by the increase level of γ H2AX (Fig. 4h). Therefore, NANP silencing independently affects both the initial induction of radiation-induced damage and repair capability, which explains the inconsistency mentioned by the reviewer (i.e., more damage despite incomplete HR)—a contradiction that would only hold if there were a sole cause-effect relationship between repair activity and damage outcome. We have included these new data in the revised manuscript and rephrased the relevant statements to more accurately reflect our conclusion.

Regarding the use of the alkaline comet assay: although our focus includes DSB repair, ionizing radiation induces both DSBs and SSBs. The alkaline comet assay was chosen to capture the full spectrum of radiation-induced DNA breaks (both SSBs and DSBs), which is relevant to understand the overall damage burden contributing to radiosensitivity, the primary focus of the study. For DSB examination, γ H2AX related assays were used.

2. There is not a clear mechanistic explanation for how NANP silencing specifically effects TNFR1 (while not impacting global sialylation), making it important to validate this finding more fully. Given the extensive use of NANP silencing, a rescue experiment with NANP re-expression should be performed. Experiments investigating TNFR1 localization upon NANP suppression and determining the dependence of radiosensitization on the sialyltransferase ST6Gal-I would significantly improve the validity of the findings, again given that it is not clear mechanistically how NANP is having this specific effect.

RESPONSE: We agree that clarifying the specific effect of NANP silencing on TNFR1 while sparing global sialylation requires much more future investigation, which has been discussed (line 425). As suggested, we have performed a rescue experiment by NANP re-expression in NANP-silenced cells, and the results demonstrate that reintroducing NANP reverses the changes in RT sensitivity, NF- κ B signaling, mesenchymal state (Fig. 3h; Supplementary Fig. 8c; 12b-c). These data confirm that the observed effects are indeed NANP-dependent.

Regarding TNFR1 localization, multiple studies reported sialylation of TNFR1 affects its internalization thus promotes NF- κ B activation (PMID: 39615678; PMID: 29233887). Given that NANP affects TNFR1 sialylation, we performed an internalization assay to investigate TNFR1 localization upon NANP suppression. The results (Fig. 6f and Supplementary Fig. 15b) revealed increased TNFR1 internalization in NANP-silenced cells, as indicated by the greater left-shift in density plot of surface TNFR1, measured by the surface ligand binding, upon temperature shift to induce internalization. Moreover, in the presence of TNF α , NF- κ B signaling induction and the nuclear p65 level was attenuated in NANP deficient cells (Fig. 6g-h and Supplementary Fig.13c), indicating an impaired NF- κ B signaling. Together, these data suggest NANP promotes TNFR1 sialylation and thereby regulates its internalization, and finally affects NF- κ B activation.

Regarding ST6Gal-I, we note that its TPM/RPKM values in GSCs are extremely low ($TPM_{GSC20}=0.071$, $RPKM_{GSC20}=0.028$; $TPM_{GSC11}=0.089$, $RPKM_{GSC11}=0.033$), well below the thresholds for active transcription ($TPM \geq 2$ or $RPKM \geq 1$; PMID: 23615947, PMID: 21654674). Consistent with this, ST6Gal-I protein is undetectable in these cells (data not shown), making it unlikely to mediate NANP's effects in GSCs. While investigating the precise mechanism that what sialyltransferase, neuraminidase or something else mediated NANP's specific impact on TNFR1 sialylation would be of interest, it lies beyond the primary focus of the current study. In fact, although a series of studies has been published (PMID: 29233887; PMID: 39615678; PMID: 21930713; PMID: 29191829), the mechanism of how ST6GAL1 has specific effects on TNFR1 sialylation remains unclear.

Minor points:

1. RT doses shown in Figs. 1 and 8 are different than those described in the methods.

RESPONSE: Our sincere apologies for the typos introduced while using old method templates. We have corrected the RT doses in the methods as shown in line 679 of the revised manuscript.

2. In Supplemental Fig. 3 and 4, NANP protein levels via western blotting should be shown upon shRNA knockdown in GSC20 and GSC11.

RESPONSE: Thank you. Western blotting results showing NANP protein levels upon knockdown in GSC20 and GSC11 have been included as Supplemental Figure 4b and Supplemental Figure 5b, respectively.

3. In Fig. 5D, there is minimal NANP knockdown with shNANP3 despite similar effects on RT sensitivity (Fig. 4b). This should be addressed. As per major point 2 above, the specificity of NANP knockdown should be confirmed via rescue experiment.

RESPONSE: The specificity of NANP knockdown has been confirmed via both knockout and rescue experiments (Fig. 3 f-h; Supplementary Fig. 8 a-c; Supplementary Fig. 12 a-c).

4. The conclusion in lines 187-189 is confusing and doesn't seem to agree with data in Supplemental Fig. 6.

RESPONSE: Our apologies for the confusion caused by the inclusion of two factors (time and dosage); we have now rephrased the statement to separate the comparisons between time and dosage. Please refer to lines 189-191.

5. In Fig. 7E, are there other sialylation target proteins that could be probed as controls to show the specificity of the decrease in TNFR1 pulldown?

RESPONSE: Thank you for the suggestion. We have probed another sialylation target protein, PDPN, as a control to show the specificity of the decrease in TNFR1 pulldown (Fig. 6e).

6. Most of the experiments are performed in GSC20 with some reproduced in GSC11. It is noted that GSC20 resembles the mesenchymal subtype. What about GSC11? Will these results be limited to mesenchymal type tumors? Overall, the findings would benefit from reproduction in a wider array of models.

RESPONSE: We appreciate the reviewer's valuable suggestion regarding the generalizability of our findings across GBM subtypes.

GBM subtypes were determined by single-sample gene set enrichment analysis (ssGSEA) using subtype signatures derived from TCGA data and our previous studies (PMID: 20129251; PMID: 28697342). Thus, a GBM classified as one subtype means that the subtype signature expression is dominant, but can retain relatively lower activities of other subtypes' signature expression. GSC11 is classified as the classical (CL) subtype. While the CL subtype is defined by dominant enrichment of classical signatures, it may retain partial expression of other subtype signatures due to intrinsic subtype plasticity—a phenomenon supported by studies highlighting the heterogeneity and dynamic transitions between GBM subtypes (PMID: 34087162).

Notably, our key findings, including NANP silencing-induced radiosensitization and the underlying mechanism involving TNFR1-NF- κ B signaling, were consistently reproduced in GSC11 (Fig. 4d-g, 5b, 6b, 7c and Supplementary Fig. 5a-e, 9b). These results confirm that NANP targeting can radiosensitize both mesenchymal (GSC20) and classical (GSC11) subtypes.

We agree that expanding the models would further strengthen generalizability. However, the conservation of the NANP-TNFR1 axis across GSC11 (CL) and GSC20 (mesenchymal) supports that our findings are not restricted to the mesenchymal subtype. We have clarified and discussed this point in the manuscript (lines 405-409).

Reviewer #4 (Remarks to the Author):

In this manuscript, the authors conducted a CRISPR knockout screen to identify radiosensitizing targets in glioblastoma (GBM). They further investigated the molecular mechanism of one target gene, NANP. Overall, this is an interesting study; however, I have the following comments:

Major comments:

The authors used CRISPR knockout screening to identify radiosensitizing targets but validated NANP using only shRNAs. At least some key experiments should be confirmed using NANP sgRNA knockout. For the in vivo radiotherapy (RT) treatment experiments, at least two different glioma stem cell (GSC) lines should be used.

The levels of P65 protein, its phosphorylation, and its nuclear localization should be assessed in the NANP knockdown cells.

RESPONSE: Thank you for suggesting areas to strengthen our manuscript. Suggested experiments to address your concerns were performed as described below.

1. Some key experiments were validated using NANP sgRNA knockout (Fig. 3f-g; Supplementary Fig. 8a-b, 12a). Moreover, rescue experiments (Fig. 3h; Supplementary Fig. 8c, 12b-c) was additionally carried out to confirm the findings.
2. We have performed in vivo RT treatment experiments using an additional GSC, the results were included as Fig. 7c.
3. The levels of P65 protein and its phosphorylation in the NANP knockdown cells was shown in Supplementary Fig. 13a, and validated in Supplementary Figure 12c. P65 nuclear localization results has been included as Supplementary Fig. 13c.

Minor comments:

Scale bars should be included in Figures 5A and 6G.

The cutoff for high versus low NANP expression should be indicated in Figures 8E and 8F.

Can NANP be targeted by any small molecule inhibitors?

RESPONSE: Scale bars have been added to the revised figures. The cutoff for high versus low NANP expression was its median expression, which has been described in the figure legend of the revised manuscript. A NANP inhibitor was published by a group from Bristol-Myers Squibb (BMS) in 2013 (PMID: 23747226); however, this compound is not commercially available nor available from BMS.

POINT-BY-POINT RESPONSE

We sincerely thank all reviewers for their careful evaluation, and please see our responses below.

REVIEWER COMMENTS

Reviewer #1 (Remarks to the Author):

The authors have adequately addressed all prior concerns from this reviewer.

RESPONSE: Thank you for helping us to improve our manuscript.

Reviewer #2 (Remarks to the Author):

The authors have expanded the number of experiments to confirm and respond to the questions of the reviewers, which is highly appreciated. Knockout of NANP in the GSC20 cell line was performed and 2 clones were selected (1 and 5) with clear NANP protein decrease on western blot (supplemental Figure 8a). Experimental details are very limited on their generation. Clone 1 was used to confirm the effect on RT, while the rescue by wildtype transfection was done again on the shRNA line. It is unclear why clone 5 was shown but not tested for effect on RT and why the wildtype rescue was not performed on the knockout clones (as a more convincing model).

The causal link of the claimed effect of TNFR1 specific desialylation on RT has been addressed by an IP of TNFR1, followed by SNA lectin staining and the results section has been adapted on page 8. In panel 1 of supplemental Figure 15a, I don't see convincing evidence of a single band that was stained with SNA in lanes 3-4, or even a band with lower intensity in the shN1 lane (4) as compared to the shC lane (3). For RCA (second panel), 2 bands are specifically stained that disappear after PNGaseF, which is as expected. This may indicate that a significant portion of TNFR1 is already desialylated in normal conditions, but at least should increase in intensity upon NANP knock-down. What is worrying is that the band of IB TNFR1 (panel 3) shows increased migration in the gel (lower MW) upon PNGaseF treatment (lanes 3-4 versus lanes 5-6; as expected because entire N-glycans are removed), but the silver stained bands (panel 4) in lanes 3-6 move at exactly the same height, suggesting the same MW. In my view, you would also expect a lower MW in the silver stained bands upon PNGaseF treatment. Another question is why these experiments were done in the shRNA model, which is more complicated than the now available KO model.

All in all, the added data (especially knockouts) increase evidence of the effect of NANP on RT, but in my view these data are not convincing to support a role of NANP in TNFR1-specific sialylation and this claim should in my view be omitted from title and paper. Possibly, NANP has additional functions or otherwise results in the clear effect of NANP on RT, which may be discussed.

RESPONSE: We sincerely appreciate your valuable feedback, which has helped us strengthen the rigor of our study. Below are our responses addressing your concerns, with a focus on key evidence and scientific rationale:

(1) Regarding the knockout clones:

The detailed protocol for generating NANP knockout clones has been added to the revised manuscript (Lines 461–467). Similar to Clone 1, Clone 5 was also used to confirm NANP's effect on radiosensitivity via clonogenic assays (in **original Figure 3f**) and new added cell cycle analyses (**Supplementary Figure S8**).

The decision to perform rescue experiments in knockdown cells rather than knockout clones was based on two considerations. **First**, CRISPR-mediated gene knockout relies on frameshift mutations introduced through non-homologous end joining (NHEJ)-mediated repair of DNA double-strand breaks. Consequently, only a subset of cells in a population will harbor such mutations, making single-cell clone isolation a necessary and standard practice (Nature Protocol, PMID: 35197604; STAR Protocol, PMID: 38972040). In our experiments, five independent single clones were selected, but only two (Clones 1 and 5) showed complete NANP knockout as validated by western blot. Given the inherent heterogeneity of tumors and the prolonged culture period (approximately four months from single cell isolation to clonal expansion), clonal variability in the generated knockout lines is unavoidable. Rescue experiments in individual clones might therefore be inherently biased toward the specific genetic background of each clone, limiting generalizability. In contrast, single-clone selection and expansion is not necessary for gene knockdown using shRNA, which avoids clone-specific variability. **Second**, as detailed in the Methods section, establishing knockout clones is extremely time-intensive, particularly for patient-derived glioblastoma spheroid cells. Generating these clones required almost half year (encompassing sgRNA construction, viral transduction, blasticidin selection, knock out validation at bulk cell level, single-clone isolation, clone expansion, and western blot validation for each clonal line). To avoid delaying the revision, we performed parallel rescue experiments in knockdown cells instead of waiting for knockout clone-based rescues.

We fully acknowledge the importance of excluding off-target effects, as emphasized by all the four reviewers. Our findings originated from a CRISPR knockout screen with four different sgRNAs and have been validated using 4 distinct shRNAs in some experiments and 2 shRNAs in numerous others across GSCs and cell lines. In previous revision, we have added further validation using an sgRNA distinct from those in the original CRISPR library, alongside rescue experiments. These data, we believe, adequately address concerns about off-target effects—an assessment supported by the other three reviewers who initially raised this issue.

(2) Regarding TNFR1 bands migration upon PNGase F treatment (supplemental Figure 15a):

We agree with the reviewer that the TNFR1 bands should expect a lower MW upon PNGaseF treatment, which was exactly the case in our data. As shown in **Figure A** below (bottom panels with alignment lines added for clarity), the bands in lanes 5–6 are clearly lower than those in lanes 3–4. As a control, the light chain bands in the same experiment align perfectly. We have marked this band shift with an asterisk (*) in previous revision. To show this more clearly, we have now replaced this data with a stronger exposed version (**Figure A** left panel) as new supplemental Figure 15a. The reason that the migration difference of proteins appears more pronounced in the western blot than in the silver staining was that different gel gradients were used. For Western blot (WB) analysis, proteins were resolved on 4–

15% gradient gels to optimize the resolution of the target band (25-75 kDa). For silver staining, broader gradient (4–20%) gel was used to separate a wider range of proteins for evaluating the purity and success of immunoprecipitation. In Summary, treatment with PNGase F resulted in increased migration (i.e., a lower apparent molecular weight) of the TNFR band in both images, consistent with the expected reduction in molecular weight following deglycosylation.

Supplemental Figure 15a
with dashed lines

Supplemental Figure 15a
with dashed lines (stronger
exposure version)

Figure A Supplemental Figure 15a of our previous revision with dashed lines to clarify the points raised by Reviewer 2. * Indicates the molecular weight for TNFR1 with or without PNGase F digestion.

(3) Regarding TNFR1 sialylation (supplemental Figure 15a):

Thank you for your thoughtful suggestion to perform target protein immunoprecipitation (IP) followed by SNA blotting as a complementary approach to validate TNFR1 glycosylation. We fully appreciate the intent to strengthen our analysis and agree that multiple orthogonal methods add rigor to glycosylation studies, however, this is technically challenging. Nevertheless, we tried our best to obtain results by this workflow by testing four distinct SNA reagents from two suppliers (**Figures B**). Two

reagents—SNA-Fluorescein (Vector Laboratories, Cat: FL-1301-2) and SNA-Biotin (Vector Laboratories, Cat: B-1305-2)—failed to detect specific bands (**Figure B panels I–III**). In contrast, the other two lectins—SNA-FITC (Thermo Fisher Scientific, Cat: L32479, Figure A) and SNA-Cy5 (Vector Laboratories, Cat: CL-1305-1, **Figure B panel IV**)—successfully detected glycosylated TNFR1 at the same molecular weight identified by the TNFR1 antibody. Both positive reagents showed reduced signals in NANP knockdown samples, supporting our conclusion. Although the signal intensity is weak, a single band stained with SNA is visible in lane 3, with an even fainter band in lane 4 (marked with an asterisk in the previous figure and boxed here for emphasis). As expected, these bands were absent following PNGase F digestion —confirming their glycosylation dependence. To show this more clearly, we have now replaced the figure with a stronger exposed version (**Figure A left panel**, new Supplemental Figure 15a). In the original supplemental Figure 15a, we presented SNA-FITC (Thermo Fisher Scientific, Cat: L32479) because, despite weaker TNFR1 bands, this reagent produced fewer non-specific signals; a stronger exposure (with enhanced contrast) is shown here (**Figure A**). SNA-Cy5 (Vector Laboratories, Cat: CL-1305-1) yielded stronger specific bands (**Figure B panel IV**, top panel); these bands (unlike non-specific bands at ~37 kDa or ~75 kDa) were eliminated by PNGase treatment, further confirming that the bands at the TNFR1 molecular weight represent immunoprecipitated glycosylated TNFR1. We have now also included SNA-Cy5 data in the revised manuscript (Supplementary Figure S15b). Given the technical difficulty of SNA blotting, many recent high-impact studies in glycosylation research only present results from SNA pull-down followed by target protein blotting, rather than target protein immunoprecipitation followed by SNA blotting (Cancer Discovery PMIDs: 37272843 [Figure 5C]; Gastroenterology 33022277 [Figure 1E]; JCI Insight 36345944 [Figure 4G]; J Biol Chem 39615678 [Figure 7E]). Through extensive efforts, we have demonstrated TNFR1 sialylation regulation using both approaches, supplemented with TNFR1 endocytosis data as indirect evidence of sialylation-dependent outcomes. We believe our data provided sufficient evidence of sialylation regulation that meets, if not exceeds, the standards of current studies in this field.

Figure B TNFR1 sialylation analysis through TNFR1 immunoprecipitation (IP) followed by SNA blotting. (I–III) SNA blotting results of TNFR1 IP with two reagents that did not detect specific bands in IP samples. M stands for marker, IP-s stands for IP supernatant. (IV) SNA blotting with SNA-Cy5. * Indicates the molecular weight for TNFR1 with or without PNGase F digestion.

Thank you for acknowledging that our RCA blotting results showed the expected bands. RCA staining did not increase following NANP knock down. As explained in our point-by-point response, the

intracellular glycosylation network is complex and incompletely understood. It is plausible that other glycosyltransferases or deglycosylases are activated when sialic acid is depleted, modifying exposed galactose residues and thus preventing a predictable increase in RCA staining. Notably, **none** of the aforementioned SNA-focused studies (PMIDs: 37272843; 33022277; 36345944; 39615678) nor more recent studies (J Clin Invest, PMID: 40371640; Nat Cell Biol, PMID: 39984654) demonstrated reciprocal regulation of SNA and RCA signals. We maintain that the data should be interpreted as observed (per your confirmation of RCA band identity).

Collectively, we have demonstrated that TNFR1 sialylation is decreased using both SNA pull-down and TNFR1 IP approaches, with PNGase F digestion to confirm specificity and silver staining to verify IP purity. Our results are consistent across two independent SNA reagents (SNA-FITC, Thermo Fisher; SNA-Cy5, Vector Labs) from different suppliers, ensuring reproducibility. Moreover, NANP's effect on TNFR1 sialylation—and the consequent impact on TNFR1 function—was further confirmed by the internalization assay shown in Figure 6f. We believe our data—with its rigorous validation, cross-reagent consistency, and clarity—provides sufficient, field-standard evidence for NANP-dependent TNFR1 sialylation, and we therefore retain this claim in the title and manuscript.

Reviewer #3 (Remarks to the Author):

The authors have addressed the comments thoroughly, and overall, the additional data supports their conclusions. There is one minor concern with the new data shown in Supplementary Fig. 12b, as two of the three MES/NF- κ B genes show minimal restoration of expression with NANP rescue. Analyzing a larger panel of genes (such as that done initially in Fig. 5f and 6g) would better indicate whether NANP rescue restores this expression phenotype as the authors conclude. There also seem to be some incorrect figure references in lines 215-226.

RESPONSE: Thank you for your recognition of our improvement. Per your suggestion, we have analyzed a larger panel of genes for NANP rescue experiments and the new added data is in new Supplementary Fig. 12b. There were **no** incorrect figure references in lines 215-216, we referenced this panel twice to describe results at two different time points (1 and 24 hours, respectively).

Reviewer #4 (Remarks to the Author):

The authors have addressed all my concern!

RESPONSE: Thank you for helping us to improve our manuscript.

POINT-BY-POINT RESPONSE

REVIEWERS' COMMENTS

Reviewer #2 (Remarks to the Author):

I would like to thank the authors for clarifying the use of different gradient gels to explain the difference in PNGaseF treated mobility of TNFR1. The use of the additional SNA-Cy5 reagent produces much cleared bands with convincing bands of IP TNFR1 stained with SNA-Cy5. This indicates that TNFR1 is partly sialylated in this model in the control situation. If I am correct, lanes 3-4 should show a lower intensity SNA staining for the shN versus shC in Figure S15b. This may seem the case. In combination with the lower capturing (IP) of TNFR1 after SNA staining, I feel that the combined results at least suggest a slight reduction in sialylation of TNFR1 and can agree with the conclusions drawn in the paper.

RESPONSE: Thank you for your careful evaluation and constructive feedback, which are very helpful for improving our manuscript. We are grateful for your positive recognition of our TNFR1 sialylation analysis and its agreement with our conclusions.